# A Review of Early Injection Strategy in Premixed Combustion Engines

**Xingyu Liang** [1],*[ID]**, Zhiwei Zheng** [1]**, Hongsheng Zhang** [1]**, Yuesen Wang** [2] **and Hanzhengnan Yu** [3]

1 State Key Laboratory of Engines, Tianjin University, Tianjin 300072, China
2 Sloan Automotive Laboratory, Mechanical Engineering, MIT, Cambridge, MA 02139, USA
3 China Automotive Technology & Research Center Co. Ltd, Tianjin 300300, China
* Correspondence: lxy@tju.edu.cn; Tel.: +86-22-27406781-8017

**Abstract:** Due to the increasing awareness of environmental protection, limitations on exhaust emissions of diesel engines have become increasingly stringent. This challenges diesel engine manufacturers to find a new balance between engine performance and emissions. Advanced combustion modes for diesel engines, such as homogeneous charge compression ignition (HCCI) and premixed charge compression ignition (PCCI), which can simultaneously reduce exhaust emissions and substantially improve thermal efficiency, have drawn increasing attention. In order to allow enough time to prepare the homogeneous mixture, the early injection strategy has been utilized widely in HCCI and PCCI diesel engines. This paper is aimed at providing a comprehensive review of the effects of early injection parameters on the performance and emissions of HCCI and PCCI engines fueled by both diesel and alternative fuels. Various early injection parameters, including injection pressure, injection timing, and injection angle, are discussed. In addition, the effect of the blending ratio of alternative fuels is also summarized. Every change in parameters has its own advantages and disadvantages, which are explained in detail in order to help researchers choose the best early injection parameters for HCCI and PCCI engines.

**Keywords:** early injection; injection pressure; injection timing; injection angle; alternative fuel; blending ratio

## 1. Introduction

Due to the advantages of better fuel economy, durability, reliability, and high specific power output compared to gasoline engines, diesel engines are widely used not only for heavy-duty vehicles such as trucks, construction machines, and generators, but also for light-duty ones including passenger cars [1–3].

However, the well-known trade-off law of nitrogen oxides (NOx) and soot emissions for conventional diesel combustion makes it difficult to reduce NOx and soot simultaneously while maintaining a high level of thermal efficiency [4]. In addition, the exhaust emitted from diesel engines, especially soot emission, has been proved to have undesirable effects on human health [5–10]. To solve the problem, many advanced combustion technologies have been studied. HCCI and PCCI combustion have proven to be promising strategies.

### 1.1. Conception of HCCI and PCCI Combustion

Figure 1 shows an illustrative ɸ-T diagram with conventional, HCCI, and PCCI combustion [11]; in the figure, ɸ means equivalence ratio and T means temperature.

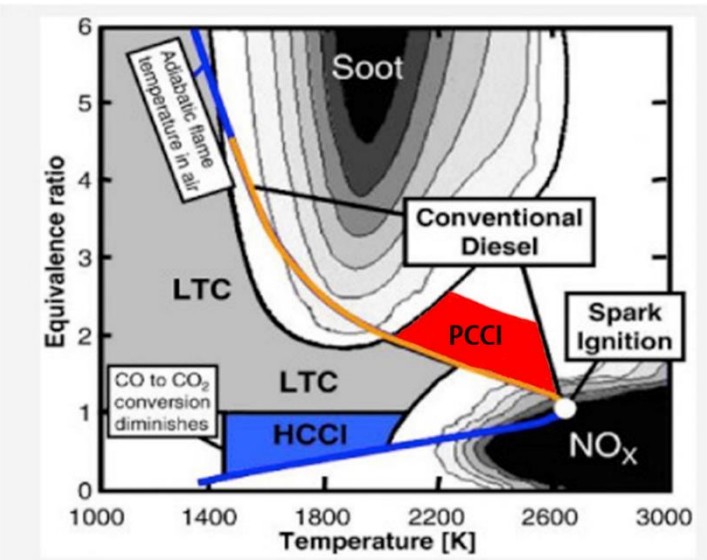

**Figure 1.** ϕ-T diagram of conventional combustion, homogeneous charge compression ignition (HCCI) combustion, and premixed charge compression ignition (PCCI) combustion [11]. ϕ, equivalence ratio; T, temperature.

The conventional diesel combustion process can be classified into four major phases: ignition delay, premixed combustion, mixing controlled combustion, and the late burning phase [12]. During the premixed combustion phase, poly aromatic hydrocarbons, the precursors of soot, are quickly formed in the hot (1600–2000 K), fuel-rich combustion regions. Soot formation follows, filling the entire downstream jet cross-section. Near the peak heat release rate of premixed combustion, a diffusion flame forms in the periphery of the fuel-rich, high-temperature downstream regions of the jet. NOx emission forms in the hot (1800–2000 K) and near-stoichiometric mixtures in the periphery of the jet near the diffusion flame. So the conventional combustion phase regime encompasses both NOx and soot islands.

HCCI combustion was first proposed by Onishi et al. [13] and Noguchi et al. [14]. The main characteristic of HCCI is a (more or less homogeneous) premixed air–fuel mixture that undergoes auto-ignition as a result of compression. The auto-ignition allows the combustion of a very lean mixture, which helps to eliminate the fuel-rich region, resulting in low soot emission. The combustion temperature is significantly lower than that of conventional diesel combustion, which is beneficial for the reduction of NOx emissions. However, a major difficulty in HCCI is to get a homogeneous admixture of air and fuel. This is especially true for diesel engines, because the lower volatility of diesel fuels makes it more difficult to obtain a homogeneous mixture compared to gasoline fuels. Besides, the high cetane number of conventional diesel fuel results in large rates of pressure rise and difficulties in combustion phasing control [15–19].

PCCI combustion has been described as a middle path between conventional and HCCI combustion modes [20–23]. In HCCI combustion, there is the challenge of combustion phasing control and homogeneous mixture preparation. To overcome these problems, for PCCI combustion, only part of the fuel undergoes the HCCI type of clean combustion, while the remainder undergoes conventional combustion. Since the remaining fuel undergoes conventional combustion, the combustion phasing is still controlled by the injection timing. Also, only partial fuel is used to prepare the homogeneous mixture, so the mixture preparation for PCCI combustion is simpler than for HCCI combustion. The part of the premixed fuel results in peak equivalence ratios staying below the soot formation threshold. Further, high levels of EGR are often used to decrease the oxygen concentration and lower peak flame temperatures, resulting in movement of the NOx island.

A comparison of the key characteristics of conventional diesel, HCCI, and PCCI combustion are presented in Table 1.

**Table 1.** Comparison of key characteristics of conventional diesel, HCCI, and PCCI combustion.

|  | Conventional Combustion | HCCI Combustion | PCCI Combustion |
|---|---|---|---|
| Injection strategy | injection close to Top Dead Center (TDC) | Early injection | Early injection + TDC injection |
| Combustion mode | Diffusion | Premixed | Premixed + diffusion |
| Ignition | Auto-ignition (controlled by injection timing) | Auto-ignition (controlled by chemical kinetics) | Auto-ignition (controlled by injection timing) |
| Combustion temperature | Partially high | Relatively low | Relatively low |
| NOx | High NOx emissions due to high combustion temperature | Low NOx emissions due to low combustion temperature | Low NOx emissions due to low temperature and exhaust gas recycling (EGR) dilution |
| Soot | High soot emissions due to diffusion combustion mode | Low soot emissions due to lean homogeneous charge | Low soot emissions due to lean homogeneous charge |

## 1.2. Early Injection Strategy

The preparation of a homogeneous mixture is important for both HCCI and PCCI combustion. In order to allow enough time for fuel to mix with the air before combustion, the early injection strategy, by which the fuel is injected in an early stage of the compression stroke, has been applied widely in HCCI and PCCI diesel engines. The start of early injection is typically 20–200° before top dead center(BTDC). Based on the characteristic of HCCI and PCCI combustion, the early injection strategy can be classified as single injection and two-stage injection, as seen in Figure 2. For two-stage injection, the first injection is also called the pilot injection, and the second injection is also called the main injection. Based on the injection timing, the early injection strategy can be divided into three patterns, as seen in Figure 3: The injection closest to TDC is defined as late; that farthest from TDC is defined as early; and the one in between is defined as middle [24]. The demarcation points of these three patterns in this paper are defined as 60°, 40°, and 20° BTDC, respectively.

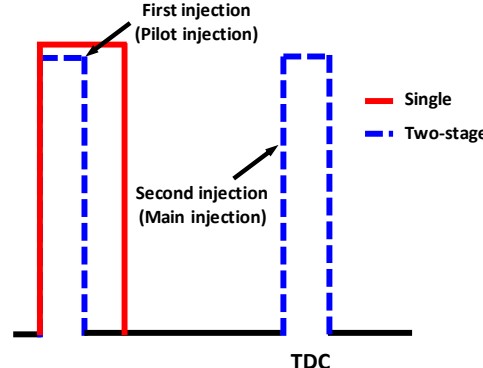

**Figure 2.** Single and two-stage early injection strategy.

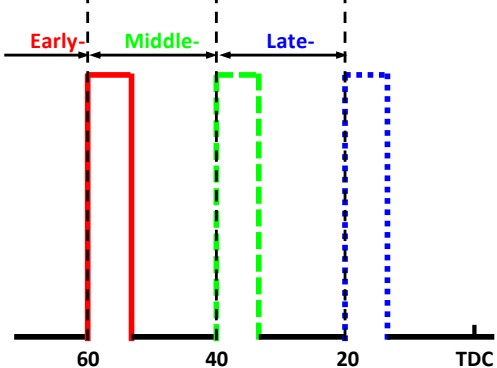

**Figure 3.** Early injection strategy divided by injection timing.

Most of these early injection strategies have a certain unavoidable influence on the fuel spray formation inside the combustion chamber, further affecting the combustion and emissions of HCCI and PCCI engines. In addition, using the early injection strategy will cause a wall-wetting problem. Because of the lower temperature of gas and the density in the cylinder during the early injection period, the fuel spray will impinge on the cylinder wall or piston head due to the slow fuel vaporization rate and longer liquid penetration length. Wall-wetting mainly leads to (1) low combustion efficiency, (2) excessive soot/carbon monoxide (CO)/hydrocarbon (HC) emissions, and (3) (local) oil dilution [25,26]. Many methods, including limiting the injection angle, have been proposed to limit or reduce wall-wetting.

*1.3. Alternative Fuel*

Changing the fuel properties and using alternative fuel are also promising ways to improve the combustion and emissions of HCCI and PCCI engines [27–29]. Biodiesel fuel, as one alternative diesel fuel, is currently of great interest and an important research subject. Biodiesel fuels contain oxygen and thus provide an effective way to eliminate the over-rich regions and enhance the combustion process, resulting in low soot, HC, and CO emissions [30–33]. Dimethyl ether (DME) is another alternative fuel. Its good ignition capability and high latent heat lead to decreased cylinder temperature in the combustion phase [34,35]. Besides, the oxygenated molecular structure and good atomization properties help in the formation of a leaner and more homogeneous mixture. The alternative fuels bioethanol and n-butanol are also widely used due to their high oxygen concentration [36–40].

As HCCI combustion is mainly controlled by chemical kinetics, the combustion process and burning rate are dependent on fuel properties. Studies have shown that optimal physicochemical properties are needed under different operating conditions; e.g., fuel with a high cetane number is required for light loads and high-octane fuel for heavy loads [41–44]. Gasoline/diesel dual-fuel combustion was proved to be a useful approach to control the combustion phasing and heat release rate of HCCI by adjusting the blending ratio according to different operating conditions [45,46].

Though many researchers have used different early injection strategies in order to reduce emissions and improve the performance of HCCI and PCCI engines, there is no comprehensive review on the effects of early injection parameters in the literature. From this point of view, the purpose of this paper is to review the effects of various parameters of early injection, including injection pressure, injection timing, injection angle, and blending ratios of alternative fuel on the performance and emissions of HCCI and PCCI engines.

## 2. Injection Pressure

The injection pressure has a direct effect on the performance and emission formation of HCCI and PCCI engines. On the one hand, higher injection pressure results in shorter injection duration and longer premixing time before the onset combustion. The increased injection pressure leads to better atomization of the fuel, which causes better air–fuel mixing and fewer fuel-rich regions. This results in a higher heat release peak, with a rapid burn rate and a shorter combustion duration, which is beneficial for engine thermal efficiency. In addition, the in-cylinder temperature for higher injection pressure is higher than that for low injection pressure. The better air–fuel mixing and higher in-cylinder temperature benefit the oxidation of soot, CO, and HC.

On the other hand, the length of spray penetration increases under higher injection pressure. This can cause serious spray-wall impingement, because both the in-cylinder temperature and pressure are low for the early injection duration. Fuel impingement on the piston head or cylinder wall leads to incomplete fuel vaporization and oxidation, which creates either over-rich or over-lean regions. This conflicts with the advantage of atomization with higher injection pressure mentioned above. This paradoxical effect finally determines the emission levels of soot, CO, and HC.

For NOx emissions, the effect of injection pressure is complex. Increased injection pressure raises the in-cylinder temperature, resulting in increased NOx emissions. On the contrary, higher injection pressure means a more homogeneous mixture, which is beneficial for realizing the low temperature

HCCI combustion mode and reducing NOx emissions. So a higher injection pressure is better for NOx reduction for more advanced injection timing.

Siewert [47] studied the effects of five injection pressures (800, 1000, 1200, 1400, and 1600 bar) on the emissions and thermal efficiency of a single-cylinder diesel engine. Results showed that with injection pressure from 800 to 1600 bar, smoke mass was progressively lowered to near zero values and HC and CO emissions were increased. Shimazaki et al. [48] studied the effect of injection pressure on the performance and emission characteristics of a premixed diesel combustion engine. The injection pressure was set from 300 to 1200 bar in increments of 300 bar. The increased injection pressure caused better NOx emissions, while CO and HC increased, especially when the injection timing was close to TDC. The fuel consumption decreased first and then increased with increasing injection pressure. Kiplimo et al. [49] investigated the influence of injection pressure on performance and emissions of a PCCI diesel engine. Two injection pressures, 800 bar and 1400 bar, were studied. The results showed that the higher injection pressure led to higher in-cylinder pressure, a higher heat release peak, and shorter combustion duration due to better atomization. The indicated thermal efficiency and indicated mean effective pressure (IMEP) were also higher with higher injection pressure. Soot and HC emissions decreased, while NOx emissions increased with increased injection pressure due to better atomization and higher in-cylinder temperature. Jeong et al. [50] investigated the effect of injection pressure on the combustion and emission characteristics of a direct injection (DI) diesel engine using a two-stage combustion strategy. The injection pressure was changed from 500 bar to 900 bar. Results showed that there were low NOx emissions, even though the injection pressure was increased. In addition, as the injection pressure increased, soot emissions were steadily reduced due to the better mixing between air and fuel. Park et al. [51] also reported the emission characteristics of bioethanol-blended diesel fuel at early injection with different injection pressures (400 and 1200 bar). Results showed that soot decreased obviously with the increased injection pressure, while HC and CO emissions increased, especially for more advanced injection timing. For NOx emissions, the effect of injection pressure was complex; higher injection pressure was better for NOx reduction for more advanced injection timing. Fang et al. [52] studied the effects of different injection pressures (600 and 1000 bar) on the combustion process in a high-speed direct-injection (HSDI) optical diesel engine employing the early injection strategy. They found that increasing the injection pressure resulted in higher NOx emissions because of the leaner air–fuel mixture and higher in-cylinder temperature. Spray development and interaction with the piston with different injection pressures were also investigated. Ayush et al. studied [53] the effects of injection pressure on a PCCI engine. The results showed that with increased injection pressure, the emissions of NOx and particulate matter (PM) declined, and the lowest point was at 700 bar. This was due to lower in-cylinder temperature. Figure 4 shows the heat release rate (HRR) variations. The increased injection pressure caused rising HRR and reduced peak in-cylinder pressure due to better air–fuel mixing. Similar results were obtained by Abraham [54] using numerical simulation methods; the simulation results showed that increasing the pressure led to increased mixing and decreased fuel-rich regions. Liu et al. [55] showed that there was little effect on ignition delay and soot peak as the injection pressure was changed from 600 to 1400 bar. Although the higher injection pressure benefited fuel atomization and mixing, it also led to more fuel wall-wetting under early injection conditions. Chen et al. [56] investigated the effect of injection pressure on the emissions and performance of a diesel engine using the two-stage injection strategy. The range of injection pressure was 1000 to 1400 bar. Results showed that IMEP and NOx emission increased, while soot decreased with the increased injection pressure because of better atomization. Arun et al. [57] reported the effect of injection pressure in a DI engine running on carbon black–water–diesel emulsion. The nozzle opening was set at 200, 220, and 240 bar. The authors compared engine performance and emission differences at different injection pressures. Results showed that different injection pressures changed the fuel atomizing characteristics; when the injection pressure was 220 bar, CO, HC, and NOx were the least, and smoke had the same characteristics. Nanthagopal et al. [58] studied the effect of Calophyllum inophyllum methyl ester on diesel engine performance, emissions, and combustion

characteristics at different injection pressures. Their investigations with injection pressures of 200, 220, and 240 bar were carried out to analyze parameters including brake thermal efficiency (BTE), specific fuel consumption, heat release rate, and engine emissions of a direct injection diesel engine fueled with 100% biodiesel. The results showed that brake-specific fuel consumption (BSFC) decreased with increased injection pressure; meanwhile, HC, CO, and smoke opacity decreased, but NOx increased. It was also found that the maximum BTE of Calophyllum inophyllum methylester was obtained at 220 bar injection pressure during part load operation conditions because the droplet size of the fuel decreased. Deep et al. [59] compared the effects of different injection pressures (200, 250, and 300 bar) on single-cylinder compression ignition (CI) engines fueled with a 20% blend of castor biodiesel in diesel. They found that under the same conditions of injection timing, as injection pressure increased, CO, HC, and smoke opacity increased but NOx emissions were significantly reduced. However, the changes of BSFC and BTE were uncertain. Jagannath et al. [60] investigated the influence of injection pressure (200, 225, and 250 bar) on the performance and emissions of a diesel engine using biodiesel blended with diesel as the fuel. Results showed that as the injection pressure increased, the brake thermal efficiency of the engine gradually increased and BSFC decreased. The reason was improved spray characteristics, better atomization, and good mixing with air at higher injection pressures.

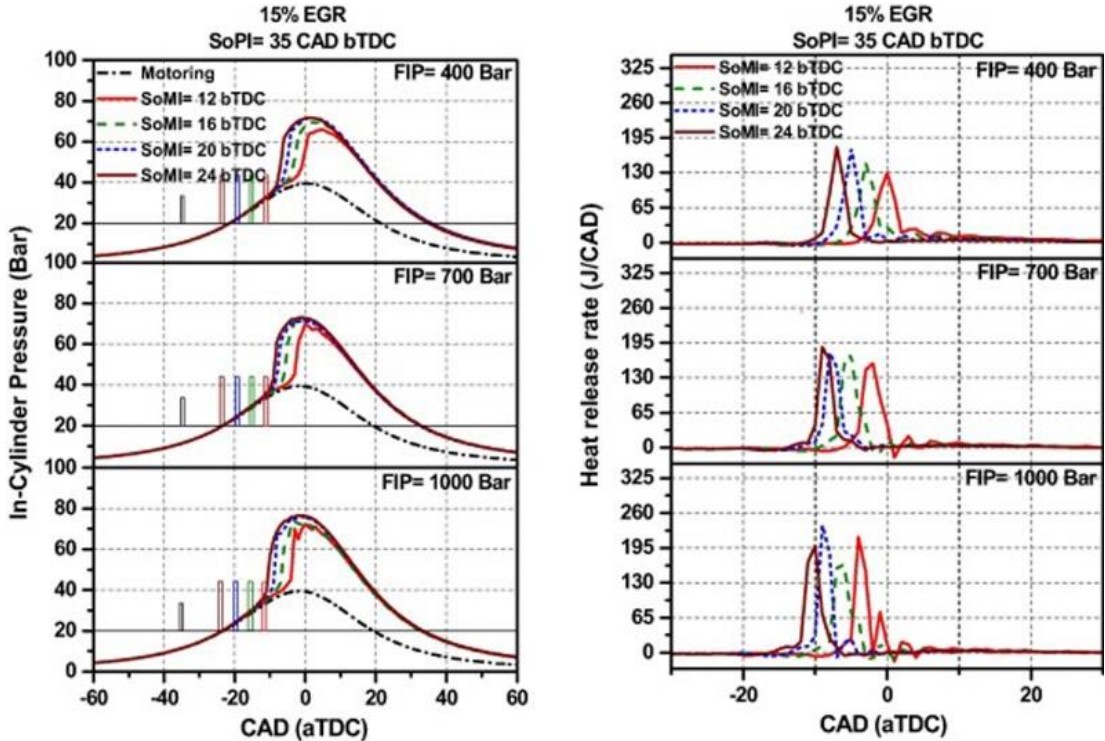

**Figure 4.** In-cylinder pressure and HRR variations with crank angle at different fuel injection pressures (FIP) [53].

Table 2 shows a summary of the variations of performance BSFC and emissions (NOx, HC, CO, and soot) of HCCI and PCCI engines after increasing the injection pressure. It can be found that for both engines, with increased injection pressure, the engine thermal efficiency improved. NOx emissions increased slightly due to the higher combustion temperature. The levels of soot, HC, and CO emissions were determined by the paradoxical effect of better atomization or more serious wall impingement. However, soot emissions were always reduced by increasing the injection pressure.

**Table 2.** Variation of performance and emissions after increasing the injection pressure. BSFC, brake-specific fuel consumption.

| Author | Injection Pressure (bar) | Fuel | BSFC | NOx | HC | CO | Soot |
|---|---|---|---|---|---|---|---|
| Jeong et al. [50] | 500–900 | Diesel | na | → | na | na | ↓ |
| Fang et al. [52] | 600/1000 | Diesel | na | ↑ | na | na | ↓ |
| Shimazaki et al. [48] | 300–1200 | Diesel | ↓↑ | ↓ | ↑ | ↑ | → |
| Kiplimo et al. [49] | 800/1400 | Diesel | ↓ | ↑ | ↓ | → | ↓ |
| Liu et al. [55] | 600–1400 | Diesel | na | na | na | na | → |
| Chen et al. [56] | 1000–1400 | Diesel | ↓ | ↑ | na | na | ↓ |
| Siewert [47] | 800–1600 | Diesel | ↓ | na | ↓ | ↓ | ↓ |
| Park et al. [51] | 400/1200 | Bioethanol blends | ↓ | ↑ | ↑ | ↑ | ↓ |
| Arun et al. [57] | 200–240 | Carbon black–water–diesel | na | ↓↑ | ↓↑ | ↓↑ | ↓↑ |
| Nanthagopal et al. [58] | 200–240 | Biodiesel | ↓ | ↑ | ↓ | ↓ | ↓ |

↑, increase; ↓, decrease; ↓↑, first decrease then increase; ↑↓, first increase then decrease; →, no significant effect; na, data not available.

## 3. Injection Timing

### 3.1. Single Early Injection

Injection timing plays a significant role in the combustion process and emissions of a diesel engine, especially when the early injection strategy is utilized. As mentioned in Section 1.2, the early injection strategy can be classified into early, middle, and late injection based on the timing.

On the one hand, injecting the fuel at an earlier time prolongs the ignition delay and helps to create a more homogeneous mixture. The formed lean mixture is then burned at a low temperature, resulting in low NOx emissions.

On the other hand, the cylinder pressure and temperature are low under earlier injection timing, which leads to poor fuel evaporation and the wall-wetting problem. Under earlier injection conditions, the fuel–air mixture is mostly formed at the outside of the combustion chamber, and some local rich mixture regions are formed due to the wall-wetting issue. Moreover, the negative work during the compression stroke increases because of the earlier combustion event. These all deteriorate the combustion efficiency and increase the products of incomplete combustion. For HC and CO emissions, the impingement target is an important factor. For middle injection timing, spray wall impingement occurs on the piston head or the outside part of the combustion chamber. For early injection timing, the impingement occurs on the cylinder wall. The different temperature and flow motions of the piston head and cylinder wall directly affect the evaporation process of the wall film. In addition, research results show that when the impingement target is at the bowl–lip area, the fuel–air mixing can be better and low HC and CO emissions can be achieved.

For soot emission, the factor of injection timing has two opposite effects. Earlier injection timing means that longer premixing allows the mixture to reach a lower equivalence ratio for low temperature, which restrains the generation of soot. Moreover, a longer time for soot oxidation is also achieved with earlier injection. On the contrary, spray wall-wetting will form some local rich regions, especially in the crevices, which promotes soot generation. Furthermore, the temperature of these wall-wetting regions is generally lower, which prevents the oxidation of soot emission.

Benajes et al. [61] investigated the influence of injection timing on particle emissions with early fuel injection timing of low-temperature diesel combustion. The injection timing was set from 33 to 24° BTDC. Results showed that PM mass and particle number increased with advanced fuel injection timing; the number of particles larger than 50 nm was especially increased. This was mainly because the higher relative levels of liquid fuel deposition on the piston bowl surface formed a locally rich

mixture, which promoted soot generation. HC and CO emissions were also increased due to the spray overshoot. Kiplimo et al. [49] studied the impact of injection timing on the performance and emissions of an HCCI diesel engine. The results indicated thermal efficiency and IMEP decreased with advanced injection timing. Injection timing earlier than 30° BTDC resulted in higher smoke emissions. This could be affected by the fuel impinging on the piston surface and splashing to the crevices, causing a rich-fuel zone. NOx emissions were lower with earlier injection timing. With the injection timing advanced, CO and HC emissions increased dramatically, owing to the wall-wetting. Kim and Lee [62] examined the influence of injection timing on the performance and NOx emissions of an HCCI diesel engine. Results showed that IMEP decreased rapidly as the injection timing was advanced beyond 20° BTDC. When the injection timing was set between 30° and 50° BTDC, IMEP was approximately half of that of conventional diesel combustion. NOx emissions were strongly affected by the injection timing. As the injection timing was advanced beyond 30° BTDC, NOx emissions fell near to nearly 0. Kim et al. [63,64] investigated the effect of injection timing on the characteristics of mixture formation and combustion in an HCCI engine. The injection timing was set from 90° to 40° BTDC. Results showed that in the case of early injection, the spray directly impinged upon the wall, a rich mixture was formed near the wall, and a lean mixture was formed in the center of the combustion chamber and the piston bowl zone, as seen in Figure 5. This led to incomplete combustion and decreased IMEP. On the other hand, earlier injection timing means a longer time for soot oxidation, which is beneficial for soot reduction. NOx emissions were also decreased with earlier injection timing. Miyamoto et al. [65] modeled the combustion process in a direct injection compression ignition engine employing very early injection timing from 180° to 20° BTDC. The results showed that the fuel injection timing influenced the start timing of heat release, further affecting the temperature distribution before and after combustion. The overall lean fuel mixture made by the extremely early injection restrained the increased in-cylinder temperature, resulting in low NOx emissions. However, the spray wall impingement caused by early injection timing increased the unburned fuel emission, such as HC emission. Kook et al. [66] investigated the effect of injection timing on premixing and combustion in a single-cylinder diesel engine. The injection timing was varied from 50° to 200° BTDC. The results showed that the early injection produced negative work. This negative work decreased as the injection timing was advanced by 70° BTDC due to the extended ignition delay period and retarded combustion phasing, and this caused increased IMEP and thermal efficiency. However, at more advanced injection timing, IMEP and thermal efficiency started to decrease because of the decreased combustion temperature. Decreasing trends in NOx emissions with advancing injection timing were seen. Smoke emission decreased at advanced injection timing due to the lean mixture and low flame temperature. In contrast to smoke and NOx emissions, HC and CO emissions increased at advanced injection timing. Figure 6 shows an image of the spray formation process with different injecting timings. Fang et al. [67] investigated spray and combustion in an optical diesel engine using a narrow-angle injector with the early injection strategy, with timing set at 40°, 60°, and 80° BTDC. Results showed that for all early injection timings, the heat release rates showed a typical two-stage heat release pattern. The earliest rapid pressure increase was seen with injection timing of 40° BTDC, and the latest was seen with 80° BTDC.

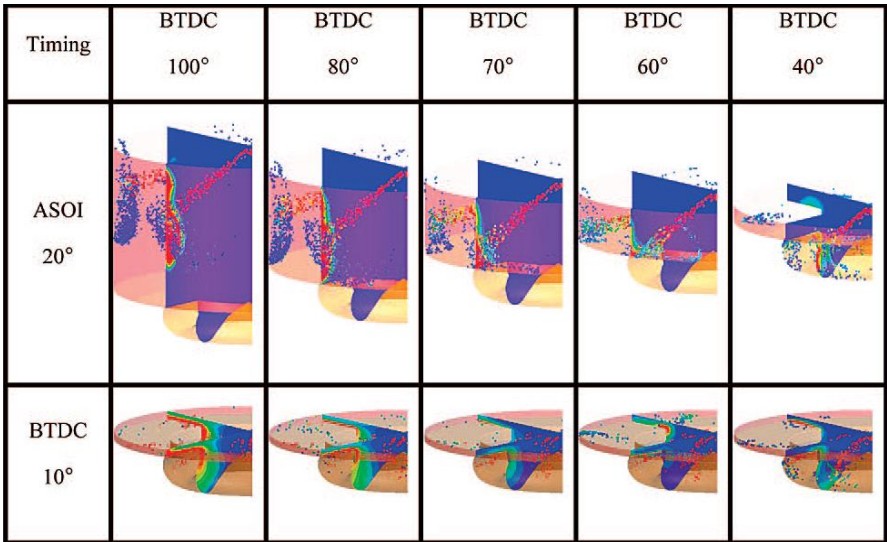

**Figure 5.** Air–fuel distribution at various injection timings [63]. Reprinted with permission from Energy & Fuels. Copyright 2019 American Chemical Society. After Start Of Injection (ASOI).

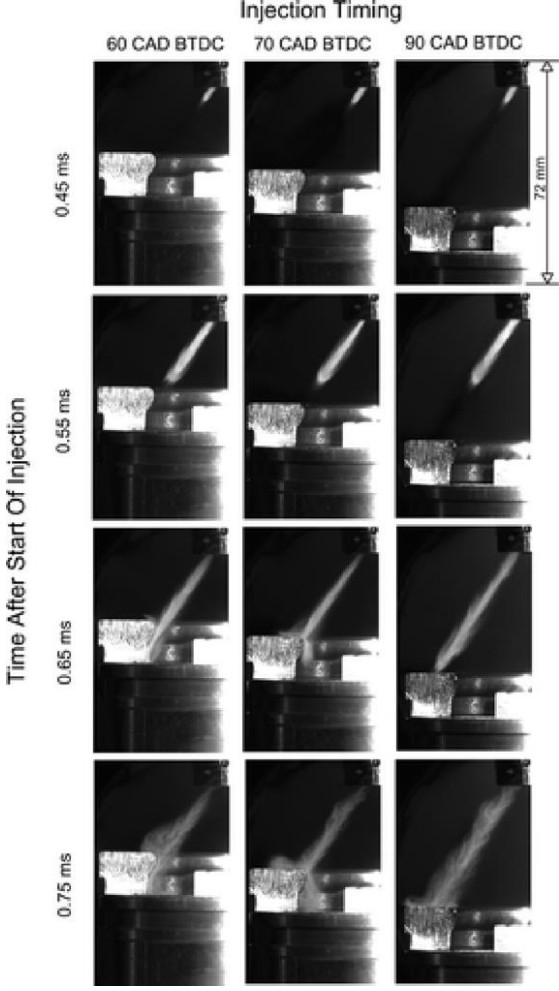

**Figure 6.** Spray development in a constant-volume vessel: injection quantity = 11.5 mm$^3$; injection pressure = 120 MPa [66]. Reprinted with permission from Energy & Fuels. Copyright 2019 American Chemical Society.

Park et al. [51] reported the emission characteristics of bioethanol-blended diesel fuel under early injection conditions. The results showed that retarding injection timing led to a decrease in ignition delay and an increase in net IMEP. NOx was reduced through advanced injection timing by the ethanol blending. This was because the injected ethanol fuel absorbed heat in the combustion chamber during the long ignition delay. Soot emission increased slightly and HC emission increased extremely with an advance in injection timing for ethanol blending due to the spray impingement. CO emission also increased because of the advanced injection timing due to the low ambient pressure, temperature, and incomplete combustion. Yoon et al. [68] investigated the effect of injection timing on DME combustion and exhaust emission characteristics. Results showed an overall decreasing trend of IMEP with advanced injection timing due to the early ignition. DME combustion indicated a low filter smoke number, and decreased with the advance of injection timing. NOx emission decreased rapidly with the advance of injection timing due to the lean premixed charge and the higher latent heat of DME evaporation. HC and CO emissions increased rapidly at more advanced injection timing due to the interaction of spray and piston head. Kim et al. [69] investigated the effect of early injection timing on combustion characteristics in a direct injection compression ignition engine fueled with gasoline. They found that IMEP decreased monotonically as the injection timing advanced due to the increased negative work. With early injection timing, HC and CO emissions increased compared with late injection timing. However, NOx emissions were reduced rapidly when the injection timing was advanced in the early injection region. This was due to the lower combustion temperature caused by lean combustion. Wamankar and Murugan [70] investigated the effect of injection timing on a direct injection (DI) diesel engine fueled with a synthetic fuel blend. Different injection timings were chosen from 20° to 26° BTDC to compare the effect on the performance and emissions of a DI engine. The results showed that the earlier the injection timing, the higher the maximum combustion temperature in the cylinder, which caused higher NOx emission; lower BSFC; and lower CO, HC, and soot emissions. Sindhu et al. [71] studied reduction of NOx emissions from diesel engines using split injection. In that paper, different injection timings were chosen; with delayed injection, the temperature of the cylinder got higher, NOx emission increased, and soot decreased. Deep et al. [59] studied the influence of injection timing on the performance and emissions of a single-cylinder CI engine fueled with a 20% blend of castor biodiesel in diesel. Results showed that different injection timings had different effects on the engine. Rakopoulos et al. [72] also found that with earlier injection timing, soot decreased and NO increased. Jagannath et al. [60] reported the effect of injection timing (24°, 27°, and 30° BTDC) on performance and smoke emissions of a CI engine. The results showed that with an injection timing of 27° BTDC, BTE was the highest and BSFC was the lowest. As for smoke opacity, the lowest value appeared at 30° BTDC due to advancement of injection timing, with extended ignition and decreased charge temperature and pressure. In research by Parka et al. [73], according to changes of injection timing in a single-injection combustion engine, it was found that NOx tended to increase with advanced injection timing, and HC and CO emissions were the lowest at an injection timing of 15° BTDC, then increased again because of combustion instability. Meanwhile, with injection timing of 15° BTDC, IMEP and brake mean effective pressure (BMEP) were highest, which is considered the most favorable combustion condition. Damodharan et al. [74] reported on the combined influence of injection timing (21°, 23°, 25° BTDC) and EGR on combustion, performance, and emissions of a DI diesel engine fueled with neat waste plastic oil. The results showed that at all EGR rates, when the injection timing was 21 crank angle degree (CAD) BTDC to 25 CAD BTDC, NOx emissions increased because of the higher combustion temperature and CO, HC, and smoke density decreased due to the higher oxygen content. BSFC decreased because there was ample time for fuel–air mixing. BTE was higher when the injection timing was earlier at all EGR rates.

Table 3 shows a summary of the variation of performance and emissions of the HCCI engine after advancing the early injection timing. In general, advancing the injection timing results in better NOx emissions but worse HC and CO emissions. Soot emission depends on the opposite effects mentioned

above. Engine performance deteriorates with advanced injection timing due to the increased negative work and incomplete combustion.

**Table 3.** Variation of performance and emissions after advancing the early injection timing (single).

| Author | Injection Timing (° BTDC) | Fuel | BSFC | NOx | HC | CO | Soot |
|---|---|---|---|---|---|---|---|
| Benajes et al. [61] | 33–24 | Diesel | na | na | ↑ | ↑ | ↑ |
| Kiplimo et al. [49] | 40–20 | Diesel | ↑ | ↓ | ↑ | ↑ | ↑ |
| Kim and Lee [62] | 70–20 | Diesel | ↑ | ↓ | na | na | na |
| Fang et al. [67] | 80–40 | Diesel | na | ↓ | na | na | ↑ |
| Kim et al. [63] | 180–20 | Diesel | ↑ | na | na | na | ↓ |
| Kim et al. [64] | 180–20 | Diesel | ↑ | ↓ | na | na | ↓ |
| Miyamoto et al. [65] | 180–20 | Diesel | na | ↓ | ↑ | na | ↓ |
| Kook et al. [66] | 200–50 | Diesel | ↑ | ↓ | ↑ | ↑ | ↓ |
| Park et al. [51] | 40–20 | Bioethanol blends | ↑ | ↓ | ↑ | ↑ | ↑ |
| Yoon et al. [68] | 40–20 | DME | ↑ | ↓ | ↑ | ↑ | → |
| Kim et al. [69] | 40–20 | Gasoline | ↑ | ↓ | ↑ | ↑ | na |
| Wamankar and Murugan [70] | 26–20 | Diesel | ↑ | ↓ | ↑ | ↑ | ↑ |

*3.2. Two-Stage Early Injection*

In PCCI combustion, a two-stage early injection strategy is utilized: Part of the fuel is first injected into the cylinder to form a homogeneous mixture prior to ignition, and this part undergoes the HCCI type of clean combustion. The remainder of the fuel is injected close to the TDC to control the combustion phase, and this part undergoes conventional combustion.

3.2.1. First Injection Timing

With the advance of first injection timing, the combustion mode transitions from conventional diffusion combustion to premixed HCCI combustion. Earlier first injection timing causes a more homogeneous in-cylinder mixture at the time of ignition since enough time is available to attain a large part of the premixed mixture, and the combustion process is divided into two stages, as seen in Figure 7. The first stage is HCCI combustion mode, which contains low-temperature reaction (LTR) and high-temperature reaction (HTR); the second stage begins just after the start of the second injection with the diffusive combustion because of the short ignition delay. With the advance of the first injection, the interval between the first and second stage combustion is first prolonged and then shortened. This is because with a too-early first injection, the fuel might inject into clearance and squish regions where the temperature is relatively low, and the mixture in these regions is over-lean and hard to be oxidized, resulting in a longer ignition delay.

Based on the different combustion processes mentioned above, the effects of first injection timing on engine performance and emissions are discussed. Variations of BSFC of first injection timing mainly depend on the weight of the heat release process. When the first injection timing is closer to TDC, BSFC is improved. With advanced first injection timing, the combustion event shifts to the earlier side, which increases negative work, and wall-wetting becomes more serious, which causes incomplete combustion. For NOx emissions, advancing the first injection timing induces better fuel–air mixing and reduces the local over-rich phenomenon, resulting in lower combustion temperature, which decreases NOx emissions. However, the intense and fast combustion caused by premixed combustion leads to increased NOx. For soot emission, better fuel–air mixing due to earlier first injection restrains soot generation and relatively higher in-cylinder temperature in the expansion stroke because the second injection promotes the oxidation of soot emission. So the soot emission generally decreases with the advance of first injection timing. HC and CO emissions commonly increase with the advance of first injection timing because there is more fuel in crevices and wall-wetting, resulting in incomplete combustion near the cylinder wall.

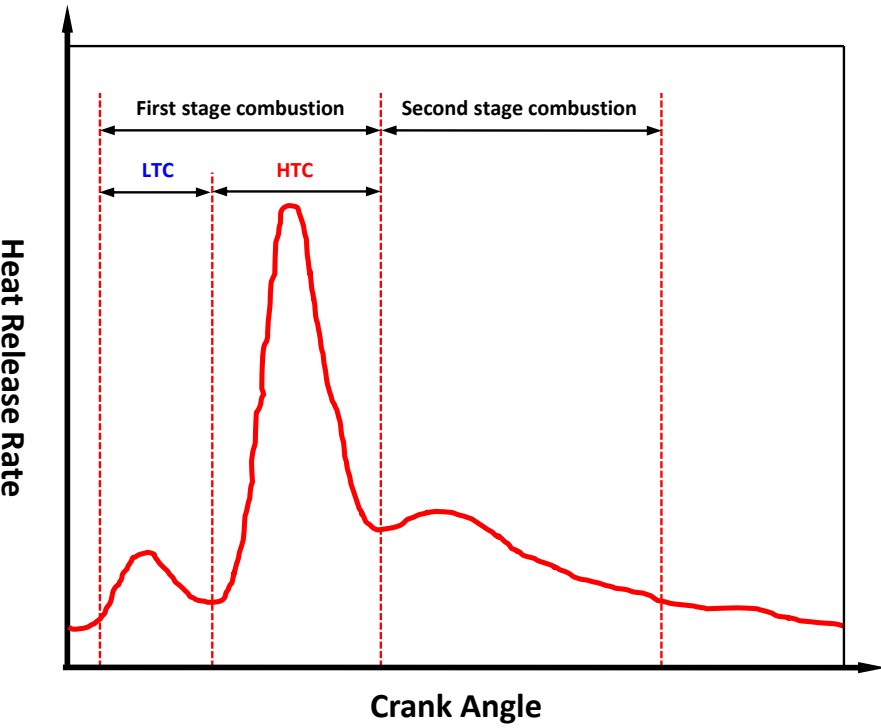

**Figure 7.** Representative heat release characteristics of PCCI combustion. L/HTC: low-/high-temperature combustion.

Mobasheri and Peng [75] computationally investigated the operating impact of first injection timing on the performance and emissions of a heavy-duty diesel engine. The first injection timing was swept from 30° to 15° BTDC, and the second injection was set at 9° BTDC. The results showed that more advanced first injection timing produced a more homogeneous in-cylinder mixture to attain a large part of the premixed mixture and higher heat release rate, resulting in increased NOx emissions. Moreover, for earlier injection timing, some of the spray missed the piston bowl. This phenomenon will cause the increase of BSFC. Abdullah et al. [76] investigated the effect of first injection timing on a modern V6 common rail direct-injection diesel engine. The first injection timing was changed from 30° to 21° BTDC, and the second injection timing was fixed at 1.4° after top dead center (ATDC). Results showed that earlier first injection timing affected the intermediate ignition delay, leading to a complete combustion process. As a result, the early injection timing produced higher in-cylinder pressures, causing higher temperatures and NOx emissions but lower soot emission. Torregrosa et al. [77] investigated the sensitivity of NOx and soot emissions to early injection in PCCI diesel engines. The first injection timing was changed from 34° to 26° BTDC, and the main injection timing was kept constant at 18° BTDC. Soot emission generally increased as the early injection timing was brought closer to TDC, and NOx emissions remained significantly lower than conventional diesel engines for all the first injection timing. In addition, BMEP decreased with the early injection timing. Yin et al. [78] investigated the optimization of a split injection strategy for light-vehicle diesel low-temperature combustion. First injection timing ranged from 10° to 35° BTDC, and the EGR rate and second injection timing were set at 50% and TDC, respectively. The results showed that as first injection timing was retarded, the typical NOx–soot trade-off was obtained, NOx emissions initially decreased but increased below 15° BTDC, and soot emissions increased sharply. On the contrary, BSFC decreased with the retarded first injection timing. Kim et al. [79] investigated the effect of a two-stage injection strategy on the combustion and flame characteristics of a PCCI engine. The first injection timing was changed from 70° to 45° BTDC, and the second injection timing was kept constant at 5° ATDC. The results showed that as the first injection was close to TDC, soot and NOx emissions increased, and BSFC also increased. The best performance occurred when the first injection timing was set at 60° BTDC. Jeong et al. [50] investigated

the effect of a two-stage combustion strategy on combustion and emission characteristics. The first injection timing was varied from 70° to 20° BTDC, and second injection timing was fixed at 5° ATDC. Results showed that the combustion pressure of the first stage increased up to the injection timing of 40° BTDC and decreased after that. In addition, the ignition delay became longer when the first injection timing varied from 20° to 40° BTDC, but extended after the injection timing of 40° BTDC. This is because the initial combustion of the first injection timing between 20° and 40° BTDC was rapidly generated as premixed combustion. However, the air–fuel mixture of too-early injection of fuel was ignited under low ambient temperature, and the ignition delay and combustion duration were prolonged. ISFC was increased up to 40° BTDC by the advanced injection timing. However, it was maintained at a similar value in a range from 40° to 70° BTDC of injection timing. NOx emissions were continuously decreased when first injection timing was advanced, and there was no particular effect on soot emission. CO and HC emissions were increased when injection timing was advanced due to the wall-wetting phenomenon. Yamane and Shimamoto [80] investigated the combustion and emission characteristics of direct-injection compression ignition engines by means of two-stage early fuel injection. The first injection timing was varied from 70° to 110° BTDC. Results showed that the NOx concentration was very low, less than 100 ppm, at an earlier first injection timing of 70° BTDC. NOx and soot emissions decreased as the first injection timing was advanced, but ISFC increased. In addition, the chemical species of formaldehyde (HCHO) was discussed; with the advance of first injection timing, HCHO emission increased. This was mainly because the cool flame with HCHO was quenched at the wall region with low in-cylinder temperature and not all the HCHO transitioned to hot flame for early first injection timing. Kook and Bae [81] investigated the combustion and emission characteristics of a single-cylinder PCCI engine using two-stage diesel fuel. The first injection timing was varied from 50° to 250° BTDC and the second injection timing was fixed at 20° BTDC. Results showed that extremely advanced timing of 200° and 250° BTDC showed higher IMEP values than retarded first injection timing. The highest IMEP with 200° BTDC was mainly because of the highly increased peak rate of heat release and longer ignition delay and combustion duration, which led to the main heat release stage being closer to TDC.

Yoon et al. [68] investigated the effects of a two-stage injection strategy on DME combustion and exhaust emission characteristics. The first injection timing was changed from 35° to 15° BTDC, and the second injection timing was fixed at 5° BTDC. NOx emissions were gradually reduced according to advanced first injection timing. Soot emission was similar for all first injection timings. The concentrations of HC and CO emissions were linearly and simultaneously increased as the first injection timing advanced. Yao et al. [82] experimentally investigated the effect of injection intervals on the performance and emissions of two-stage injection HD diesel fueled by n-butanol/diesel. The second injection timing was set at 5° BTDC. Soot emission decreased, while CO emission increased with the extended injection interval. In addition, the first-stage heat release became inconspicuous when the injection interval was longer than 24 CAD. Zhuang et al. [83] investigated the effects of first injection timing on combustion, performance, and emission characteristics in a DI-diesel engine fueled with diesel from direct coal liquefaction under second injection timing at 1° ATDC. The first injection timing was changed from 45.5° to 8.5° BTDC. Results showed that advancing the first injection timing advanced the start of first-stage combustion and increased the peak in-cylinder pressure, and the first-stage combustion became inconspicuous when the injection interval was longer than 25 CAD. NOx and soot emissions deceased with the advance of first injection timing. However, BSFC rose due to the earlier first injection. Zheng et al. [84] investigated the effects of first injection timing on combustion and emissions under high EGR rates in a diesel engine fueled by blends of two fuels (gasoline and n-butanol). The first injection timing was changed from 65° to 25° BTDC, and the main injection timing was kept constant at 14° BTDC. Results showed that with advanced first injection timing, the maximum heat release rate of three of four fuels (diesel, B30, and G30, but not diesel/gasoline, diesel/n-butanol (DGB)) increased. Smoke emissions decreased due to the advanced first injection timing. NOx emissions decreased and then increased slightly before and after the first injection timing

of 40° BTDC. On the contrary, CO and HC emissions increased with the advanced first injection timing. Figure 8 shows the effects of pilot injection strategies and fuel properties on in-cylinder pressure and heat release rate. Yang et al. [85] studied the effects of pilot injection timing on the combustion noise and particle emissions of a diesel/natural gas dual-fueled engine at low load. In the study, the pilot injection timing was set at 30° to 5° BTDC. The results showed that with advanced pilot injection timing from 5° to 17° BTDC, BTE increased sharply, and then a similar tendency could not be observed. In terms of emissions, with advanced pilot injection timing, brake specific HC emissions (BSHC) and brake specific CO emissions (BSCO) notably decreased, while brake specific NOx emissions (BSNOx) significantly increased because the dual-fuel combustion phase was strongly controlled by the pilot injection timing, and advanced injection timing led to an earlier combustion phase.

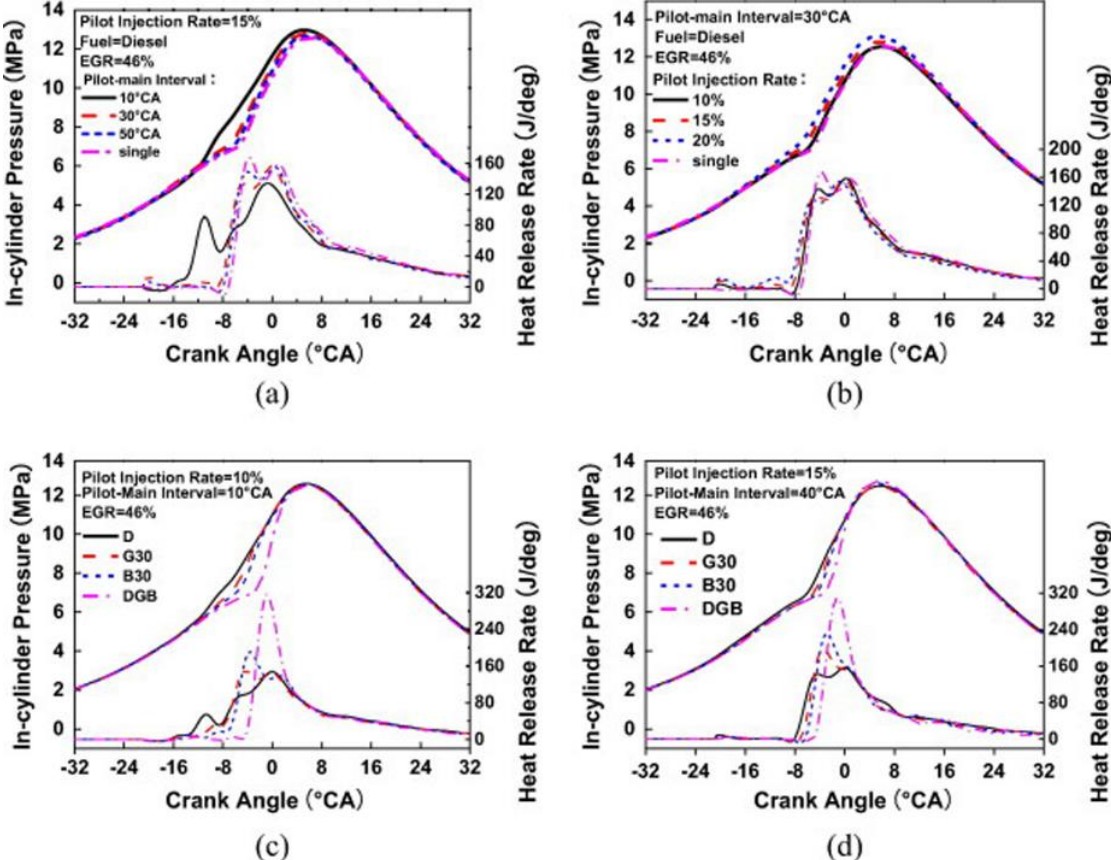

**Figure 8.** Effects of pilot injection strategies and fuel properties on in-cylinder pressure and heat release rate [84]. (**a**) pilot injection rate = 15%, different pilot-main interval, in-cylinder pressure and HRR curve with CA; (**b**)constant pilot-main interval, different pilot injection rate, in-cylinder and HRR curve with CA; (**c**) and (**d**) compares the in-cylinder pressure and heat release rate of four different fuels.

Table 4 shows a summary of the variation of performance and emissions of the PCCI engine after advancing the first injection timing. Advancing the first injection timing will generally decrease NOx and soot emissions and increase HC and CO emissions. Engine performance deteriorates with advanced injection timing due to the increased negative work and incomplete combustion.

**Table 4.** Variation of performance and emissions after advancing the first injection timing (two-stage).

| Author | First Injection Timing (° BTDC) | Second Injection Timing (° BTDC) | Fuel | BSFC | NOx | HC | CO | Soot |
|---|---|---|---|---|---|---|---|---|
| Abdullah et al. [76] | 30–9 | −1.4 | Diesel | na | ↓↑ | na | na | ↓ |
| Mobasheri and Peng [75] | 30–15 | 9 | Diesel | ↑ | ↑ | na | na | na |
| Torregrosa et al. [77] | 34–26 | 18 | Diesel | ↑ | ↓ | na | na | ↓ |
| Yin et al. [78] | 35–10 | TDC | Diesel | ↑ | ↓↑ | na | na | ↓ |
| Jeong et al. [50] | 70–20 | −5 | Diesel | ↑ | ↓ | ↑ | ↑ | → |
| Kim et al. [79] | 70–45 | −5 | Diesel | ↓ | ↓ | na | na | ↓ |
| Yamane and Shimamoto [80] | 110–70 | na | Diesel | ↑ | ↓ | na | na | ↓ |
| Kook and Bae [81] | 250–50 | 20 | Diesel | ↓ | → | → | ↑ | ↓ |
| Yoon et al. [68] | 35–15 | 5 | DME | ↑ | ↓ | ↑ | ↑ | ↓ |
| Yao et al. [82] | 42–21 | 5 | n-butanol | na | na | na | ↑ | ↓ |
| Zhuang et al. [83] | 45.5–8.5 | −1 | Diesel from Direct Coal Liquefaction(DDCL) | ↑ | ↑↓ | na | na | ↓ |
| Zheng et al. [84] | 65–25 | 14 | B30/G30/DGB | na | ↓↑ | ↑ | ↑ | ↓ |

### 3.2.2. Second Injection Timing

The second injection is considered to act as the ignition controller and promoter of PCCI combustion. The second injection timing mainly influences the second stage of the combustion process, which is mainly diffusive combustion. With retarded second injection timing, the major combustion event was delayed. The variation of BSFC of different second injection timings mainly depended on whether the combustion event shifted to near TDC. In addition, NOx emissions decreased when the second injection timing was retarded because of the low charge temperature caused by the late combustion. Soot emissions generally increased as the second injection was retarded. This was because of the increased portion of diffusion combustion and low charge temperature. HC and CO emissions also increased with retarded second injection timing. This was expected, because the charge temperature was too low to burn the second injection fuel.

Coskun et al. [21] experimentally and numerically investigated the effects of second injection variations on combustion and emissions of an HCCI-DI engine. The second injection timing was varied from 30° to 15° BTDC, and the first injection timing was fixed at 240° BTDC. Results showed that the overall combustion and emissions characteristics of the HCCI engine could be directly controlled by second fuel injection timing. Combustion pressure, maximum temperature, and NOx decreased, but HC emission increased by retarding the second injection timing. In addition, computational fluid dynamics (CFD) simulation results showed that combustion began at the fuel-rich zone between the initial homogeneous mixture and second injected fuel zone. Kook and Bae [81] investigated the combustion and emission characteristics of a single-cylinder PCCI engine using two-stage diesel fuel. The second injection timing was changed from 20° BTDC to TDC, and the first injection timing was fixed at 200° BTDC. Results showed that when the second injection timing was at TDC, two peaks of main heat release appeared, the first from the auto-ignition of premixed charge and the second because the remaining charge was ignited from the second injection. When the second injection timing was set at 15° BTDC, there was only one high peak of main heat release and its highest value could be obtained, so the combustion efficiency was improved. Torregrosa et al. [77] investigated the sensitivity of NOx and soot emissions to early injection in PCCI diesel engines. The second injection timing was varied from 26° to 8° BTDC, and the first injection timing was kept constant at 34° BTDC. They observed that the ignition delay of the second injection decreased as the second injection timing was retarded. Soot emissions generally increased as the second injection timing was brought closer to TDC, while NOx emissions decreased. Besides, BSFC decreased when the second injection was retarded. Kim et al. [79] investigated the effect of a two-stage injection strategy on the combustion and flame characteristics of a PCCI engine. With the first injection fixed at 60° BTDC, the second injection timing was varied from 5° to −7.5° BTDC. As the second injection timing was retarded, NOx emissions decreased, BSFC increased, and soot did not change much initially but increased later on. The combustion flame was distributed homogeneously in the combustion chamber, and HCCI combustion was noticeable when the second injection timing was retarded after TDC due to the increased time for more homogeneous mixture formation, as seen in Figure 9. Kim and Lee [62] investigated the effect of a dual-injection

strategy on improving exhaust emissions in an HCCI diesel engine. Second injection timing was varied from TDC to 20° ATDC under a fixed first injection timing of 60° BTDC. When the second injection was delayed, the location and shape of the first-stage heat release from the first injected fuel almost unchanged, while the second stage heat release of diffusive combustion occurred later and the heat release rate was slowed. At the retarded second injection timing, the major combustion event was shifted to later than TDC, which obtained a decrease of NOx emissions because of the low charge temperature and decreased IMEP. Total HC and CO emissions increased as the second injection timing was retarded. Kanda et al. [21] investigated the effect of second injection timing on a PCCI diesel engine. With the first injection timing fixed at 56° BTDC, the second injection timing was changed from 5° to 18° BTDC. Results showed that retarding the second injection timing shortened the ignition delay of the second injection due to the high ambient temperature, which increased the portion of diffusive combustion and resulted in significantly increased soot emissions. Retarding the second injection timing also decreased IMEP due to the shift of combustion to the late side.

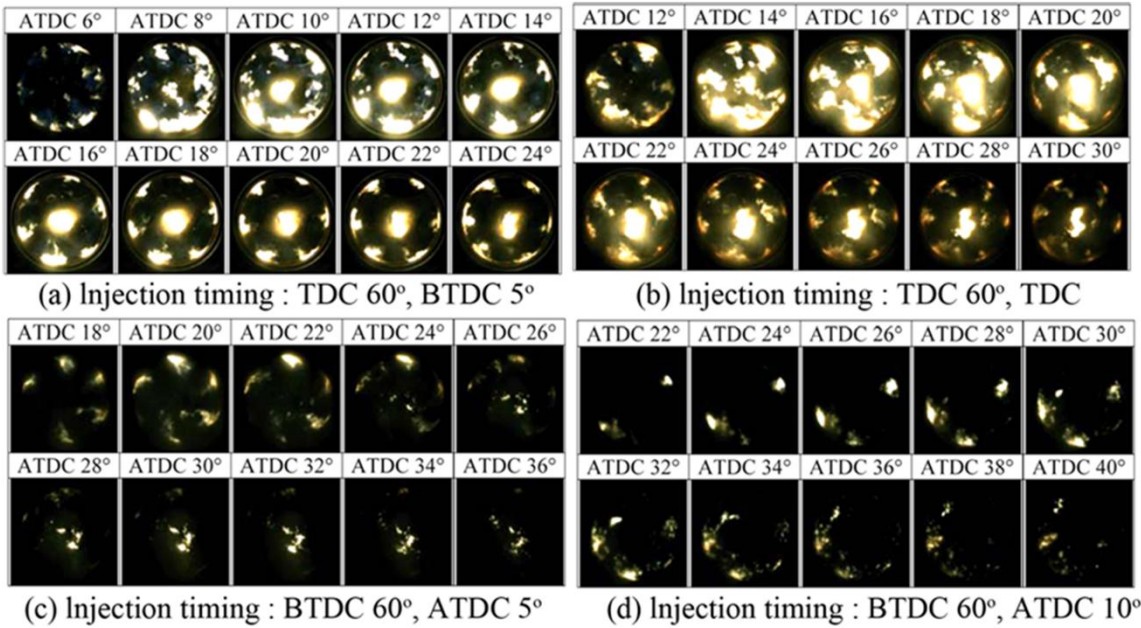

**Figure 9.** Flame characteristics as a function of second injection timing change [79]. (**a–d**): different second injection timing, the images of flame change with crank angle.

Yao et al. [82] experimentally investigated the effects of second injection timing on the performance and emissions of two-stage injection heavy duty (HD) diesel fueled by n-butanol/diesel. The first injection timing was fixed at 42° BTDC, and the second injection was varied from 10° to −2.5° BTDC. Soot emissions declined with retarded second injection timing. However, CO emissions were just the opposite. In addition, BSFC increased when the second injection timing was retarded. Yoon et al. [86] investigated the effect of second injection timing on the emissions of a DME HCCI engine. The first injection timing was fixed at 80° BTDC, and the second injection timing was varied from 20° BTDC to TDC. Results showed that HC emission increased slightly as the second injection was retarded near TDC. The retarded second injection caused the combustion to continue after TDC. As a result, the unburned fuel was increased by the lower in-cylinder temperature and pressure during the expansion stroke. CO emission also increased as the second injection was retarded because of the decreased cylinder temperature. However, retarding the second injection prolonged the time to form a leaner mixture, and it burned under low-temperature conditions during the expansion stroke, resulting in a drop of NOx emissions.

Turkcan et al. [87] investigated the effects of second injection timing on combustion and emissions of an HCCI-DI engine. Results showed that retarding the second injection timing increased the cooling

effect and inhomogeneity of the mixture, which decreased peak cylinder mean temperature values (Tmax). NOx emissions decreased, while CO emissions increased due to incomplete combustion. Das et al. [88] studied the effects of main injection timing on controlling the combustion phasing of a homogeneous charge compression ignition engine using a new dual-injection strategy. The main injection timing (MIT) was investigated from 26° to 8° BTDC. Results showed that BTE for HCCI combustion was consistently lower as the MIT was retarded. BSFC was the opposite. As the MIT was retarded, NOx emissions decreased because peak cylinder pressure got lower. As the injection timing was retarded, ignition delay of the main fuel was further reduced, resulting in increased smoke formation. Because the retarded main injection put off the combustion phasing and subsequently increased ignition delay, the pilot fuel was mixed better, leading to rapidly increasing HC emissions.

Table 5 shows a summary of the variation of the performance and emissions of the PCCI engine after retarding the second injection timing. In general, retarding the second injection timing will reduce NOx emissions, but increase soot, HC, and CO emissions. Engine performance deteriorates with retarding of second injection timing due to the shifting of diffusive combustion to later than TDC.

**Table 5.** Variation of performance and emissions after retarding the second injection timing (two-stage).

| Author | First Injection Timing (° BTDC) | Second Injection Timing (° BTDC) | Fuel | BSFC | NOx | HC | CO | Soot |
|--------|---------------------------------|----------------------------------|------|------|-----|-----|-----|------|
| Coskun et al. [89] | 240 | 30–15 | Diesel | na | ↓ | ↑ | na | na |
| Torregrosa et al. [77] | 34 | 26–8 | Diesel | ↓ | ↓ | na | na | ↑ |
| Kook and Bae [81] | 200 | 20-TDC | Diesel | ↑ | na | na | na | na |
| Kanda et al. [21] | 56 | 18–5 | Diesel | ↑ | na | na | na | ↑ |
| Kim et al. [79] | 60 | 5 to −7.5 | Diesel | → | ↓ | na | na | ↑ |
| Kim and Lee [62] | 60 | TDC to −20 | Diesel | ↑ | ↓ | ↑ | ↑ | na |
| Yoon et al. [86] | 80 | 20-TDC | DME | na | ↓ | ↑ | ↑ | na |
| Yao et al. [82] | 42 | 10 to −2.5 | n-butanol | ↑ | na | na | ↑ | ↓ |

### 3.2.3. First Injection Quantity

The first injection quality is in direct relation to the proportion of HCCI combustion and diffusive combustion. Increasing the first injection quantity caused more fuel and air to be mixed before second injection, which is beneficial for auto-ignition of first injection fuel, and more heat was released before second-stage combustion. So the ignition timing of the mixture was advanced, resulting in increased BSFC. Moreover, increasing the first injection quantity shortened the ignition delay and raised the combustion temperature of the second-stage combustion, which promoted the formation of NOx emissions. However, NOx emissions could be limited with the use of heavy EGR. Besides, a large first injection quantity might cause wall-wetting and more fuel to enter the region near the cylinder wall, which would aggravate the incomplete combustion, leading to more CO and HC emissions. For soot emissions, on the one hand, the wall-wetting caused by the first injection promoted its generation. On the other hand, the higher temperature and smaller proportion of diffusive combustion also decreased soot emission, but usually it was increased with increased first injection quantity.

Torregrosa et al. [77] investigated the effects of first injection quantity on NOx and soot emissions in PCCI diesel engines. The first injection quantity was assessed by considering between 20% and 60% of the total fuel mass. The first injection timing was fixed at 34° BTDC, and second injection timing was changed from 26° to 8° BTDC. Results showed that increasing the first injection quantity resulted in decreased engine BEMP and increased BSFC. This was because with the increased first injection quantity, the combustion phasing advanced and less fuel was burned with proper combustion phasing. NOx emissions decreased as the first injection quantity increased. Soot emissions changed slightly with the change of first injection quantity. Zheng et al. [84] investigated the effects of first injection quantity on combustion and emissions under a high EGR rate in a diesel engine. First injection quantity ratios were chosen from 10–20%. The first and second injection timings were set at 44° and 14° BTDC, respectively. Results showed that increasing the first injection quantity led to increased cylinder pressure and advanced due to more fuel being mixed with air, which resulted in better combustion performance. Increasing the first injection quantity also increased NOx, HC, and CO emissions. There

was no obvious difference in smoke emissions when the first injection rate was changed. Jeong et al. [50] investigated the effects of a two-stage combustion strategy on combustion and emission characteristics. The first injection mass was varied from 2 to 10 mg/cycle. The first and second injection timings were fixed at 50° and −5° BTDC, respectively. When the first injection quantity was increased, the peak pressure increased and the heat release of the first stage increased gradually and the ignition delay decreased. NOx first decreased then increased as the first injection mass increased. The lowest NOx emission was obtained when the first and second injection quantities were the same, and soot emission increased rapidly as the first injection quantity increased. Kim et al. [79] investigated the effects of the first injection rate in a PCCI engine. The fuel mass portion of the first injection was varied from 10% to 50% of the total fuel mass with the first and second injection timing fixed at 60° and −5° BTDC. The results showed that both the fuel mass part of the first injection and the cylinder peak pressure increased, which resulted in a shorter ignition delay for the second combustion, causing rapid combustion before TDC and increased BSFC. PM emission increased as the first injection rate increased because of the shorter ignition delay time before the second combustion occurred and the wall-wetting issue. On the contrary, NOx emissions decreased because of the lower rate of heat release (ROHR) peaks of the second combustion when the first injection quantity was increased. As shown in Figure 10, the first injection portion of 20% seemed to cause a brighter flame than that of 30%, meaning a higher heat release rate and higher peak combustion temperature. At the first injection portion of 30%, neither the rapid combustion flame nor the diffusion flame appeared in the combustion chamber owing to the uniform mixture distribution. When the first injection portion was 40% and 50%, a diffusion flame generating soot appeared due to the short ignition delay and mixing time. Lee et al. [90] investigated the effects of fuel quantity during first injection on the emissions of a DI diesel engine under moderate and heavy EGR. Under moderate EGR (~30%), the quantity of the first injection was changed from 1 to 5 mg/cycle. The first injection timing was changed from 100° to 10° BTDC and second injection timing was fixed at TDC. The gas temperature in the cylinder was reduced when the first injection quantity increased, resulting in lower NOx emissions. However, because of the early timing of the first injection, most of the first injection fuel was not included in the main combustion process. As a result, HC emissions increased as the first injection quantity increased, and BSFC deteriorated. Soot emissions were also higher as the first injection quantity increased, and an early and small first injection was helpful to reduce the soot emission. Under heavy EGR (~60%), the amount of the first injection ranged from 1 to 10 mg/cycle. The first injection timing was changed from 45° to 25° BTDC and the second injection timing was fixed at 15° BTDC. CO, HC, and PM emissions decreased as the amount of the pilot injection increased. A large pilot injection closer to the main injection was recommended. The contrary results between moderate and heavy EGR were due to the differences in ignition delay and air–fuel mixture conditions. Zhuang et al. [83] investigated the effects of first injection quantity in a DI-diesel engine fueled with diesel from direct coal liquefaction. The first injection quantity was varied from 2 to 6 mm$^3$/cycle, timing was changed from 44.5° to 7.5° BTDC, and the second injection timing was fixed at −1° BTDC. Results showed that with increased first injection quantity, the maximum heat release rate of first-stage combustion and the peak in-cylinder pressure rose; moreover, the maximum heat release rate of the second stage was reduced. These changes shortened the ignition delay of second-stage combustion. NOx emissions increased with the increment of first injection quantity. Soot emissions were slightly influenced by the first injection quantity. Neely et al. [91] experimentally investigated the effects of first injection quantity on the emissions and combustion of a PCCI diesel engine. The first injection quantity was varied from 15% to 35% of the total fuel mass, and the first and second injection timings were fixed at 38° and 4° BTDC. Results showed that the combustion temperature increased and the start of combustion advanced as the first injection quantity increased, resulting in increased NOx emissions. CO emissions also increased because the portion of wall-wetting fuel increased, causing incomplete combustion. The increased incomplete combustion also increased the BSFC. De Ojeda et al. [92] investigated the effects of first injection quantity on the emissions of a Partially Premixed Compression Ignition (PPCI) engine.

The first injection quantity was varied from 0 to 10 mg/cycle, with the first injection timing fixed at 50°
BTDC. Results showed that soot emissions decreased with the increased first injection quantity due
to the leaner homogeneous charge mixture, while HC emissions increased slightly. Chen et al. [56]
investigated the effects of first injection quantity on the emissions and performance of a diesel engine.
The first injection quantity was swept from 20% to 60%, with the first and second injection timings fixed
at 35° and −10° BTDC. Results showed that NOx and soot increased sharply after the first injection
quantity reached 40%. Park and Bae [93] investigated the effects of first injection quantity on the
combustion and emissions of a PCCI engine. The first injection quantity was varied from 0 to 50%,
and the first and second injection timings were fixed at 30° and 25° BTDC. Results showed that as
the first injection quantity increased to 30%, the peak of heat release was raised and the combustion
phase was advanced, resulting in increased NOx but decreased soot. However, the peak of heat release
was reduced and the combustion phase was retarded again when the first injection quantity exceeded
40%, resulting in decreased NOx and increased soot emission. HC and CO emissions increased with
the increased first injection quantity due to wall-wetting. Fang et al. [94] studied the influence of pilot
injection quantity on the performance and emission characteristics of an HCCI-DI combustion engine.
The pilot injection quantity was from 0 to 10 mm$^3$. Results showed that when the pilot quantity was
smaller than 8 mm$^3$, smoke emission increased with the rise of pilot quantity, and then dropped a little
when it was bigger than that. NOx emission was opposite to soot due to the trade-off between the
reduction of NOx during HCCI combustion and the increase of NOx during diffusive combustion.
HC and CO emissions increased as the pilot quantity increased, which is a major problem of HCCI
combustion due to incomplete combustion and fuel impingement. BSFC increased with increased pilot
quantity. The reason was off-phasing of the combustion process and the negative work; meanwhile,
HC and CO increased. Yu et al. [95] investigated first injection proportions in the combustion and
emission characteristics of a diesel engine. In that study, the first injection timing of the two-stage
injection strategy was fixed at 30° BTDC, and the first injection proportion varied from 10% to 90%.
The results showed that as the first injection proportion rose, indicated specific fuel consumption
(ISFC) first decreased then increased, and the lowest value was obtained at 40%. At the same time,
soot emission decreased sharply, but NOx emission was opposite to ISFC, and the turning point was
30%. Because the heat release processes of the first and second injected fuel combustion were too close,
resulting in higher combustion temperature and NOx emission, with the increased proportion of the
first injection, more injected fuel was premixed and combusted, and less second injected fuel was
combusted in the high-temperature atmosphere, resulting in lower NOx emissions.

Table 6 shows a summary of the variation of performance and emissions of the PCCI engine after
increasing the first injection quantity. In general, increasing the first injection quantity will increase
HC and CO emissions due to wall-wetting. NOx emissions also increase due to the faster heat release
rate and higher combustion temperature caused by a large portion of premixed combustion. Soot
emission is determined by the opposite effects of wall-wetting, higher combustion temperature, and
lower proportion of diffusive combustion. As an experimental value, the first injection quantity should
be limited to 40% of the total injection fuel.

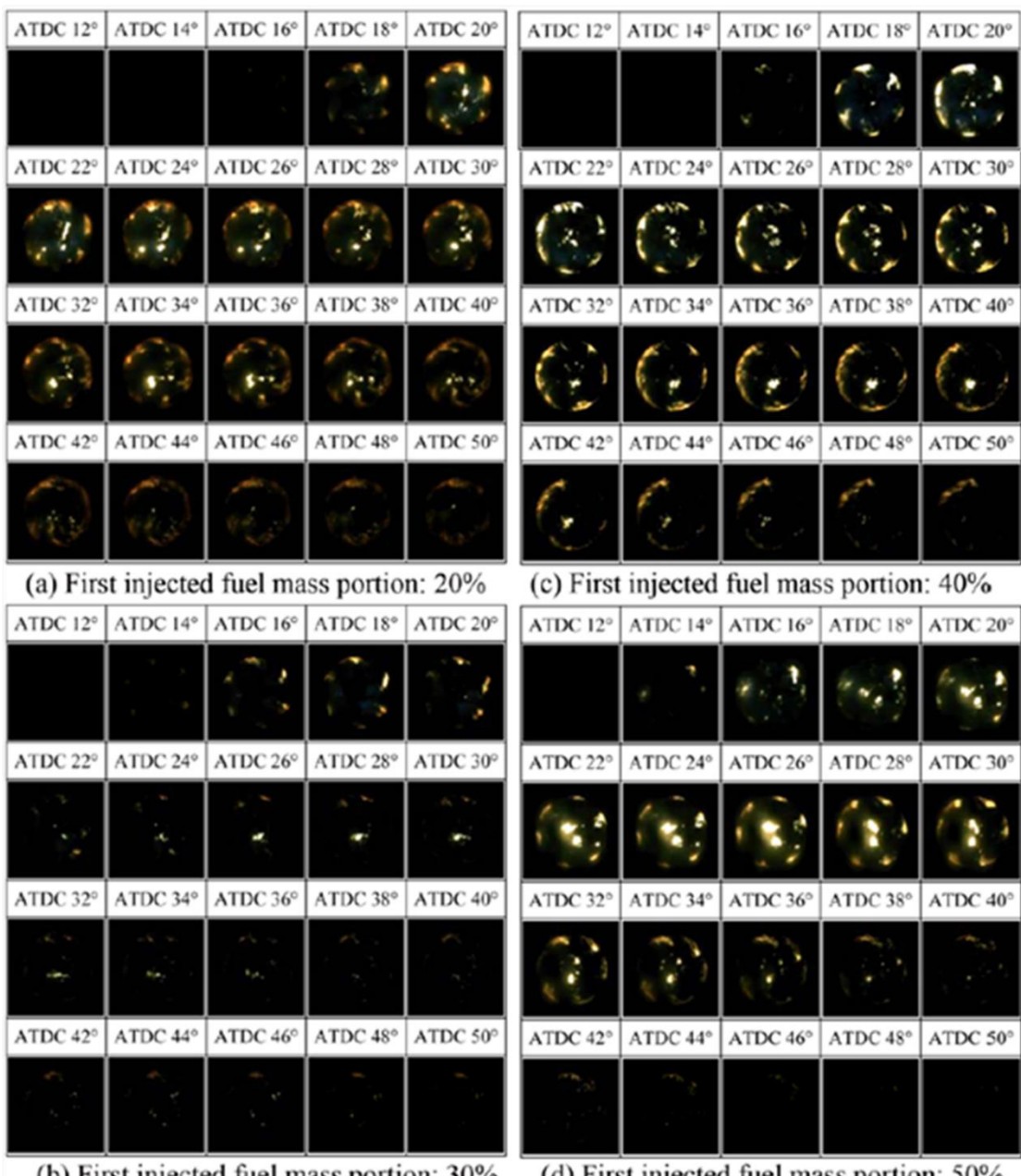

**Figure 10.** Flame characteristics as a function of the first injection quality ratio [79]. (**a**–**d**): first injected fuel mass portion: 20%, 30%, 40%, and 50%, respectively; the images of flame change with crank angle.

**Table 6.** Variation of performance and emissions after increasing the first injection quantity (two-stage).

| Author | First Injection Quantity | Fuel | BSFC | NOx | HC | CO | Soot |
|---|---|---|---|---|---|---|---|
| De Ojeda et al. [92] | 0–10 mg/cycle | Diesel | na | na | ↑ | na | ↓ |
| Lee et al. [90] | 1–5 mg/cycle | Diesel | ↑ | ↓ | ↑ | na | ↑ |
| Jeong et al. [50] | 2–10 mg/cycle | Diesel | na | ↓↑ | na | na | ↑ |
| Park and Bae [93] | 0–50% | Diesel | na | ↑↓ | ↑ | ↑ | ↓↑ |
| Zheng et al. [84] | 10–20% | Diesel | na | ↑ | ↑ | ↑ | → |
| Kim et al. [79] | 10–50% | Diesel | ↑ | ↑ | na | na | ↑ |
| Neely et al. [91] | 15–35% | Diesel | ↑ | ↑ | na | ↑ | na |
| Torregrosa et al. [77] | 20–60% | Diesel | ↑ | ↓ | na | na | → |
| Chen et al. [56] | 20–60% | Diesel | na | ↑ | na | na | ↑ |
| Zhuang et al. [83] | 2–6 mm$^3$/cycle | DDCL | na | ↑ | na | na | → |

## 4. Injection Angle

Wall-wetting caused by the early injection strategy directly influenced the performance and emissions of the HCCI diesel engine. Limiting the injection angle has been proved to be a useful approach to reduce the wall-wetting phenomenon. For the strategy with early injection timing, with a decreased injection angle, the distance between the nozzle and the cylinder wall increased relatively and more fuel was atomized before reaching the cylinder, leading to reduced wall-wetting fuel mass. For the strategy with middle injection timing, the spray with narrow injection angle was confined in the piston bowl, and the spray targeting the piston bowl varied with different injection angles. The magnitude and direction of the spray rotation in the bowl were directly affected by the injection angle, as shown in Figure 11. This difference further impacted the fuel–air mixing in the piston bowl and finally impacted combustion and emissions. As mentioned in Section 3.1, the impingement target is an important factor influencing emissions and is commonly determined by the injection timing, injection angle, and piston structure. Many researchers have shown that when the spray impinges at the bowl lip bottom edge, the secondary atomization process is enhanced, which benefits the fuel–air mixing. For the second injection of the two-stage injection strategy, using a narrow injection angle against the fuel–air mixing was proved, because the spray targeting moved to the inside part of the piston bowl, where the air motion was weak. Furthermore, the fuel film deposition of the narrow injection angle was stronger, resulting in high soot generation and incomplete combustion.

Kim and Lee [62] investigated the effects of a narrow fuel spray angle on improved exhaust emissions in a HCCI diesel engine with an early injection strategy. Two injector nozzles with different spray cone angles (156° and 60°) were used in the study. Results showed that in contrast to the conventional injector, the ISFC indicated a modest decrease when the injection timing was advanced to 50–60° BTDC in the case of a narrow angle injector. In addition, using dual-injection strategy with narrow angle fuel injection made it possible to reduce CO emissions, maintaining high thermal efficiency and low NOx emissions. Fang et al. [52] investigated the effects of injection angles on the combustion process using multiple injection strategies in an HSDI optical diesel engine. Two injector tips with different injection angles, 70° and 150°, were used. The results showed that after the first injection, the maximum cylinder pressure and heat release rate were slightly higher for 70° than 150°. However, after the second injection, the cylinder pressure peaks and heat release rate were lower for 70° than 150°. The combustion images for different injection angles and pressures at different crank angles are shown in Figure 12. Nonluminous flame was seen for the first injection of 150°, while two types of flame, nonluminous and luminous film combustion flame, could be seen for 70°. Ignition occurred near the spray tip in the vicinity of the bowl wall for the 150° tip, but it was near the injector tip in the central region of the bowl for the 70° tip. More soot luminosity was observed with the 70° tip due to fuel film combustion. On the other hand, the fuel film combustion led to lower NOx emissions due to its rich mixture. Kim et al. [63] investigated the effects of injection angle on the

characteristics of mixture formation and combustion in a PCCI engine using an early multiple injection strategy. Four injection angles (150°, 130°, 100°, and 70°) were tested. Results showed that when the spray impinged on the wall of the combustion chamber, IMEP decreased because of the incomplete combustion due to wall-wetting; IMEP increased as the injection angle decreased. As the injection angle decreased, the mass of fuel impinging on the bowl region increased and the mixture forming in the bowl region became richer, which caused increased smoke. Kook et al. [66] investigated the combustion and emission characteristics of a single-cylinder PCCI engine using two-stage diesel fuel. Two injection angles (150° and 100°) were examined. In order to reduce the wall-wetting by spray over-penetration, the injector with a narrow injection angle was tested. Results showed that opacity was improved to a minimum of 7% because the spray impingement was reduced, and HC and CO concentrations were also reduced compared to the 150° injection angle, while the power output was increased. However, with the advance of early injection timing, NOx concentration highly increased with the narrow injection angle. Mobasheri and Peng [75] computationally investigated the effects of injection angle on combustion characteristics, performance, and amount of pollutant emissions in a DI injection angle. Three injection angles (145°, 105°, and 90°) were studied. NOx emissions decreased with decreased injection angle. However, as the injection angle decreased, soot emissions first decreased and then increased, and there was a best injection angle for soot emissions, which probably depended on the interaction of the spray and the piston bowl. The trend of BSFC as the injection angle was changed was similar to the soot emissions. The increased BSFC was mainly caused by incomplete combustion of the wall film. Siewert [47] investigated the effects of varying injection angles on emissions and thermal efficiency using a DI diesel engine. Results showed that injection angles from 100° to 158° significantly changed the spray targeting the piston bowl. Soot emissions decreased sharply as the injection angle decreased. HC and CO emissions with the 100° injection angle were the minimum.

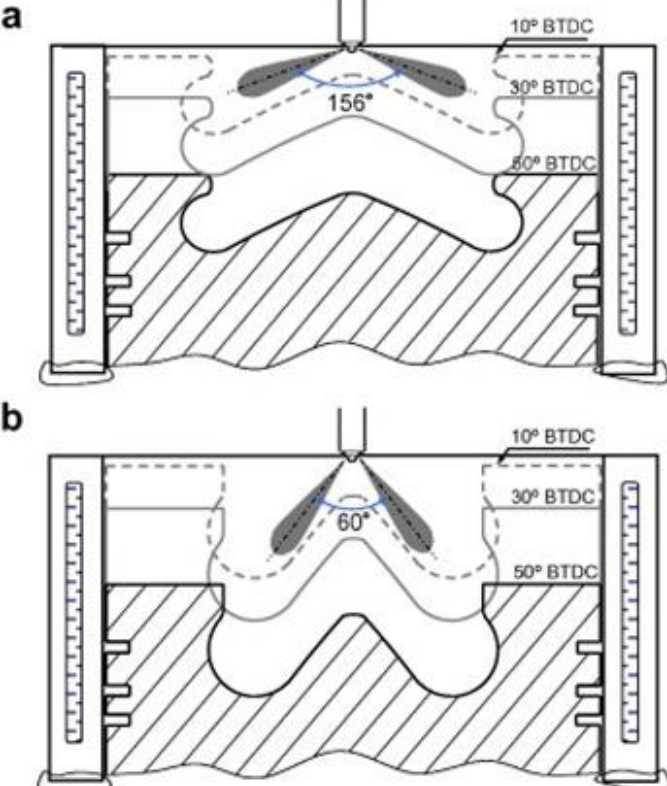

**Figure 11.** Schematic diagrams of the tested combustion chamber and fuel spray: (**a**) conventional diesel engine; (**b**) modified engine configuration for early injection [62].

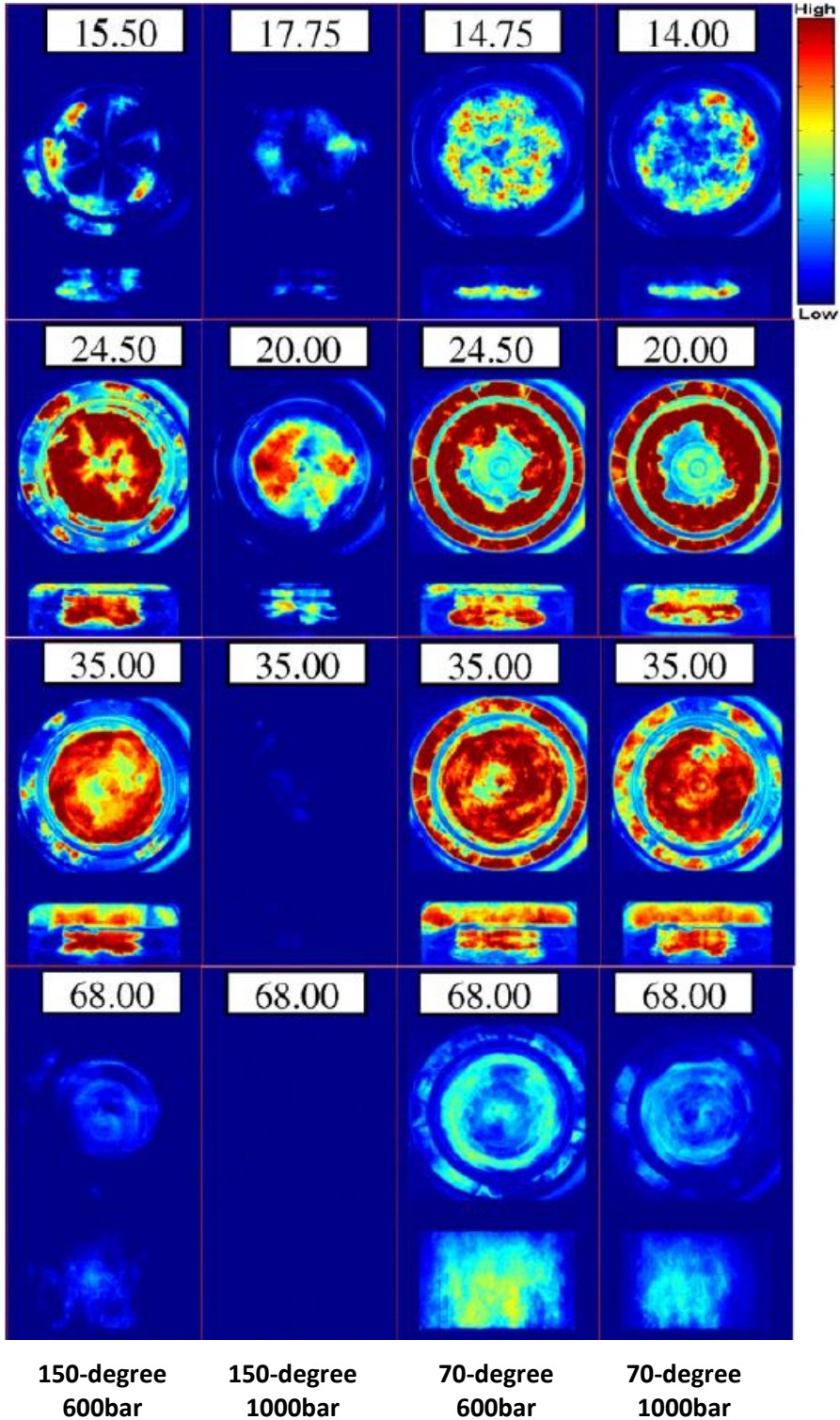

**Figure 12.** Combustion images of injection angles and injection pressures at different crank angles [52].

Park et al. [96] investigated the effects of injection angle on HC and CO emissions reduction with the early injection strategy. Two injectors with different injection angles (156° and 70°) were used. Results showed that in the combustion using the narrow angle injector, the variation in peak combustion pressure was smaller than the combustion using the conventional injector. In addition, IMEP was increased and ignition delay was shortened. The narrow angle injector induced low and stable HC and CO emissions, which was the weakness in the diesel–bioethanol blend which fueled

combustion. Yoon et al. [68] also investigated the effects of injection angle on DME combustion and exhaust emission characteristics in a common-rail diesel engine. Two narrow spray angle injectors (70° and 60°) were examined, and the results were compared with the results of a conventional injection angle (156°). For the single injection strategy, the combustion pressure and heat release rate of the narrow-angle injectors were increased as compared to the wide-angle injector. IMEP was higher for the narrow-angle injectors. Soot emission decreased slightly using narrow-angle injectors, and NOx emission stayed at a low level with all injection angles using the early injection strategy. HC and CO emissions decreased obviously using narrow-angle injectors due to the decreased wall-wetting fuel. For the multiple-injection strategy, the combustion pressure and heat release rate of the first injection with narrow injection angles were also higher than with the wider injection angle, leading to more thorough combustion. Also the ignition delay between first and second injections was shorter with the narrow injection angle. However, upon the second injection, the combustion pressure and heat release rate were lower with the narrow injection angle. The 70° injection angle showed the highest IMEP. Low soot and NOx emissions could be obtained with the narrow injection angle with earlier first injection timing. HC and CO emissions with the narrow angle were obviously better due to the reduction of wall-wetting. Vanegas et al. [97] investigated the influence of injection angle on the emissions and combustion efficiency of a PCCI diesel engine. The injection angle varied from 100° to 148°. Results showed that decreasing the injection angle was an efficient method to reduce HC emission. However, NOx emission was highest with the 100°, especially when the injection timing was earlier than 30° BTDC. In addition, smoke emission was the highest for the 100°, especially when the injection timing approached TDC. Simulation was done by Kim et al. [98] to investigate combustion and emission characteristics in a small-bore HSDI diesel engine using an injector with different injection angles. Results showed that for wider-angle injection (120° and 130°), the spray targeting played an important role in determining the combustion regime. For narrow-angle injection (50° and 85°), it was found that the flow patterns were very similar regardless of injection timing.

Table 7 shows a summary of the variation of performance and emissions of HCCI and PCCI engines after decreasing the injection angle. In general, decreasing the injection angle will limit or reduce the wall-wetting phenomenon, resulting in decreased HC and CO. However, soot emission is directly affected by the placement of spray targeting. Decreasing the injection angle generally is not good for the control of soot emission, but NOx emission can be suppressed by the rich fuel–air mixture and low combustion temperature.

**Table 7.** Variation of performance and emissions after decreasing the injection angle (two-stage).

| Author | Injection Angle (°) | Fuel | BSFC | NOx | HC | CO | Soot |
|---|---|---|---|---|---|---|---|
| Kim and Lee [62] | 60/156 | Diesel | ↓ | → | na | na | na |
| Fang et al. [52] | 70/150 | Diesel | na | ↓ | na | na | ↑ |
| Kim et al. [63] | 70–150 | Diesel | ↓ | na | na | na | ↑ |
| Mobasheri and Peng [75] | 90–145 | Diesel | ↓↑ | ↓ | na | na | ↓↑ |
| Vanegas et al. [97] | 100–148 | Diesel | na | ↑ | na | na | ↑ |
| Kook and Bae [81] | 100/150 | Diesel | na | ↑ | ↓ | ↓ | ↓ |
| Siewert [47] | 100–158 | Diesel | na | na | ↓ | ↓ | ↓ |
| Park et al. [96] | 70/156 | Bioethanol blended | ↓ | na | ↓ | ↓ | na |
| Yoon et al. [68] | 60/70/156 | DME | ↓ | → | ↓ | ↓ | → |

## 5. Blending Ratio

Because of the different physical and chemical properties of alternative fuels, the blending ratio of alternative fuel directly affects the overall spray behavior, atomization characteristics, and emission characteristics.

Fang et al. [99] investigated different blending ratios (20–100%) of biodiesel combustion in an optical HSDI diesel engine under low-load premixed combustion conditions using the early injection strategy. Results showed that fuel impingement on the wall was observed for all ratios. The liquid penetration became longer and fuel impingement was stronger with increased biodiesel ratio; also,

the ignition delay became longer and heat release curves became lower and broader. This could be explained by the higher boiling point and lower cetane number of the biodiesel. Except for B0, soot luminosity increased with increasing biodiesel ratio. This might be because the low volatility of biodiesel results in locally rich regions with higher soot luminosity, as shown in Figure 13. NOx emissions first decreased with increasing biodiesel ratio; then, after the ratio passed a certain value, NOx emissions increased. This is because of the trade-off between ignition delay and oxygen content. Park et al. [38] investigated the effects of bioethanol-blended diesel fuel on combustion and emission characteristics in an early injection diesel engine. The blending ratio of the bioethanol varied from 10% to 30%. Results showed that an increased bioethanol blending ratio extended the ignition delay due to lower cetane number and the decreased gas temperature caused by the evaporation of bioethanol with large latency. The difference in ignition delay between pure diesel and diesel–bioethanol blended fuels became larger under early injection conditions. The bioethanol–diesel blending caused a small decrease in soot emission because of the high oxygen content, and reduced NOx emission because of the low combustion temperature caused by the high heat of evaporation of bioethanol fuel and lower heating value. However, HC and CO emissions increased with an increased bioethanol blending ratio. Ma et al. [100] investigated the influence of the blended gasoline ratio on gasoline/diesel dual-fuel combustion. The blending ratio of the gasoline was increased from 68% to 84%. Results showed that a more homogeneous charge and lower global fuel reactivity resulted from increasing the gasoline ratio, which led to later combustion phasing. The retarded combustion phasing dropped the cylinder temperature, which reduced the cylinder pressure and NOx emissions. The lower cylinder temperature and more homogeneous charge resulted in decreased soot emissions. Higher HC and CO emissions observed with the increased gasoline ratio could be a combined result of increased crevice flow mass and less effective oxidization under the lower temperature. Liu et al. [55] investigated the effects of fuel properties on soot reduction under early injection conditions in a diesel engine. Results showed that as the gasoline ratio was increased from 0% to 70%, soot emissions were reduced dramatically. On the one hand, with an increased gasoline ratio, ignition delay was prolonged and fuel–air had better mixing. On the other hand, the increased gasoline proportion caused higher fuel vaporization, which shortened the spray penetration and reduced the fuel wall-wetting. In addition, the effect of n-butanol was also studied. With increasing n-butanol ratio, ignition delay increased and soot emissions decreased obviously. This was because the oxygen atoms in n-butanol could enter over-rich regions and consume soot precursors, resulting in reduced soot emissions. Yao et al. [82] also experimentally investigated the influence of the diesel fuel n-butanol blending ratio on the performance and emissions of a two-stage injection HD diesel engine fueled by n-butanol/diesel. Diesel fuels with different amounts of n-butanol (0%, 5%, 10%, and 15%) were used. Results showed that oxygenated fuel could get much lower soot and CO emissions compared to the base diesel. A higher n-butanol fraction in the blend could get lower soot and CO emissions, since the oxygenated addition was effective in fuel-rich regions. The density and cetane number of n-butanol are lower than those of diesel, therefore further increasing n-butanol in a blend will prolong injection duration and ignition delay. More fuel will burn during the expand stroke with relatively lower combustion efficiency, which will cause increased BSFC.

　　Agarwal et al. [101] reported the effects of 10%, 20%, and 50% Karanja biodiesel blends on injection rate, atomization, engine performance, emissions, and combustion characteristics of a common rail direct-injection (CRDI)-type fuel injection system evaluated in a single cylinder. The results showed that as the blend ratio increased, BSFC increased obviously. As for emissions, NOx decreased, but HC and CO increased because the higher blend ratio caused inferior mixing of fuel with air. Turkcan et al. [87] compared the effects on performance and emissions of an HCCI engine with four blend ratios of fuel (E10, E20, M10, and M20). The results showed that with increased ethanol content, Pmax decreased; meanwhile, higher methanol content meant lower maximum pressure (Pmax). In addition, the oxygen in the blended fuel and in-cylinder temperature affected by the physical properties of the additive had a direct influence on NOx and CO emissions. Guedes et al. [102] studied the performance and combustion characteristics of a compression ignition engine running on (diesel biodiesel ethanol) DBE

blends. The engine was fueled with B15E5, B15E10, and B15E15 (B, biodiesel; E, ethanol). At the same speed of torque and injection timing, results showed that BSFC increased with increased ethanol content due to the lower heating value.

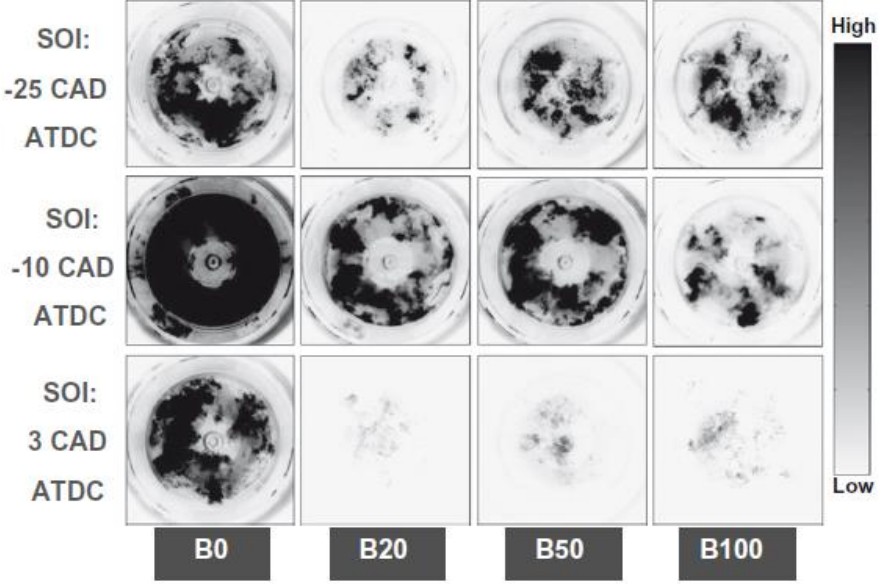

**Figure 13.** Combustion images at different crank angles under various biodiesel blending ratios [101]. SOI: start of injection.

Table 8 shows a summary of the variation of performance and emissions of HCCI and PCCI engines after increasing the blending ratio of alternative fuels. In general, for oxygenated fuel, the high heat of evaporation and lower heating value lead to reduced combustion temperature, which is beneficial for reducing NOx emissions, but is not good for the oxidation of HC, CO, and soot emissions. However, the high oxygen content of oxygenated fuel helps to consume soot precursors and HC and CO emissions in the over-rich region. Besides, the liquid penetration of the spray becomes longer with increased oxygenated because of the change of the physical properties of injected fuel, resulting in a more serious wall-wetting phenomenon. So the emission characteristics of fuel with different blending ratios is affected by the above factors. For gasoline–diesel blended fuel, the increased gasoline ratio prolongs ignition delay and promotes better fuel–air mixing. In addition, the higher fuel vaporization decreases the cylinder temperature, which results in reduced NOx and soot. However, the increased crevice flow mass and lower temperature lead to increased HC and CO, but wall-wetting is reduced because of the short penetration of the gasoline–diesel spray.

**Table 8.** Variation of performance and emissions after increasing the blending ratio of alternative fuels.

| Author | Blending Ratio (%) | Fuel | BSFC | NOx | HC | CO | Soot |
|---|---|---|---|---|---|---|---|
| Fang et al. [99] | 20–100 | Biodiesel | na | ↓↑ | na | na | ↑ |
| Park et al. [38] | 10–30 | Bioethanol | na | ↓ | ↑ | ↑ | → |
| Liu et al. [55] | 30–70 | Gasoline | na | na | na | na | ↓ |
| Ma et al. [100] | 68–84 | Gasoline | na | ↓ | ↑ | ↑ | ↓ |
| Yao et al. [82] | 0–15 | n-Butanol | ↑ | na | na | ↓ | ↓ |

## 6. Summary and Conclusions

A review was conducted of the literature concerning the effects of early injection strategies on the performance and emissions of HCCI and PCCI engines. The early injection strategies covered in this review are injection pressure, injection timing, injection angle, and blending ratio of alternative fuels.

Researchers have carried out many experimental and numerical works to investigate the effects of different injection parameters, and the most important conclusions derived are summarized below:

- Increasing the injection pressure improved the engine thermal efficiency and decreased soot emissions. This is because a higher injection pressure will result in a better fuel spray, which will make the fuel burn more completely. HC and CO emissions were determined by the paradoxical effect of high injection pressure. They are uncertain. NOx emissions increased slightly with increasing injection pressure due to the higher combustion temperature. Wetting-wall phenomenon caused by higher injection pressure can lead to the opposite results.

- Advancing the injection timing of HCCI engines was beneficial in terms of better NOx emissions, but with increased HC and CO emissions. Soot emission depended on the opposite effects of longer premixing time and more serious wall-wetting. In theory, longer premixing time can reduce soot emissions, but wall-wetting can cause incomplete or low-temperature combustion that can increase soot emissions. The engine performance deteriorated with advanced injection timing due to the increased negative work and incomplete combustion.

- Advancing the first injection timing of PCCI engines was beneficial in terms of better NOx and soot emissions, but HC and CO emissions increased and engine performance deteriorated. In addition, retarding the second injection timing also reduced NOx emissions. However, soot, HC, and CO emissions increased and engine performance deteriorated due to the shifting of diffusive combustion to later than TDC. From the previously cited literature, wetting-wall phenomenon must be considered when using two-stage early injection strategy, which has an important impact on engine performance and emissions.

- Increasing the first injection quantity of PCCI engines increased HC and CO emissions. NOx emissions also increased due to the faster heat release rate and higher combustion temperature caused by a large portion of premixed combustion. Soot emission was determined by the opposite effects of wall-wetting, higher combustion temperature, and lower proportion of diffusive combustion. As far as experimental values go, the first injection quantity should be limited to 40% of the total injection fuel, which will obtain better engine performance and emission characteristics.

- Decreasing the injection angle limited or reduced the wall-wetting phenomenon, resulted in decreased HC and CO. However, soot emission was directly affected by the location of spray targeting. Generally, decreasing the injection angle was not good for the control of soot emission, but NOx emissions could be suppressed by the rich fuel–air mixture and low combustion temperature. As the same time, the narrow injection angle is also beneficial to reduce the BSFC, due to avoiding incomplete combustion caused by wetting-wall phenomena.

- The impingement target is an important factor influencing engine emissions, and the impingement target was commonly determined by the injection timing, injection angle, and piston structure. When the spray impinged at the bowl lip bottom edge, the secondary atomization process was enhanced, resulting in a drop of HC and CO emissions.

- For HCCI and PCCI engines fueled with alternative fuels, the blending ratio of the alternative fuel directly affected the performance and emission characteristics. Oxygenated fuel was beneficial for the reduction of NOx and soot emissions, but was not good for decreasing HC and CO emissions. For gasoline–diesel blended fuel, the increased gasoline ratio resulted in reduced NOx and soot but increased HC and CO emissions.

- For HCCI and PCCI engines, an appropriate early injection strategy can effectively improve engine performance and improve engine emissions characteristics. However, due to the actual working process of the engine, its performance and emissions are affected by many factors. Therefore, it is also necessary to study the effects of multiple factors (Injection pressure; Injection Timing; Injection Angle and fuel) on engine performance and emissions.

The advantages and disadvantages of the early injection parameters have been summarized, as shown in Table 9.

**Table 9.** Advantages and disadvantages of early injection parameters.

| Early Injection Parameters | Injection Pressure ↑ | Injection Timing | | | | Injection Angle ↓ |
|---|---|---|---|---|---|---|
| | | Single ← | Two-Stage | | | |
| | | | First Injection Timing ← | Second Injection Timing → | First Injection Quality ↑ | |
| Advantage | Better air–fuel mixing and fewer fuel-rich regions. Higher heat release rate and temperature. Shorter combustion duration. Higher engine thermal efficiency. Better oxidation of soot, CO, and HC emissions. | Longer premixing time. Better air–fuel mixing and more homogeneous mixture. Lower NOx and soot emissions. | Same as advancing single early injection timing. Second combustion stage promoted soot oxidation. | Lower combustion temperature. Lower NOx emissions. | Better air–fuel mixing and more homogeneous mixture for first combustion stage. Lower NOx and soot emissions. | Decreased wall-wetting fuel. Better air–fuel mixing and more homogeneous mixture. Lower soot, CO, and HC emissions. |
| Disadvantage | Longer spray penetration. More serious wall-wetting results in more soot, CO, and HC emissions. Higher combustion temperature results in higher NOx emissions. | Lower cylinder pressure and temperature during injection period. More serious wall-wetting. Shifting combustion event to earlier side results in more negative work. Deteriorated combustion efficiency and increased incomplete combustion products. Higher soot, CO, and HC emissions. | Same as advancing single early injection timing. | Increased diffusive combustion portion. Higher soot, CO, and HC emissions. | Advancing ignition results in more negative work. Shorter ignition delay and higher combustion temperature of second combustion stage results in higher NOx emission. More serious wall-wetting Higher soot, CO, and HC emissions. | Impingement between spray and piston bowl especially for second injection. Increasing fuel deposition on piston bowl results in higher soot, CO, and HC emissions. |

### 7. Future Research Directions

From the review of the published work on the early injection parameters of HCCI and PCCI engines, the following interesting topics are identified for investigation in future research:

- Methods to limit or avoid the wall-wetting problem caused by early injection strategy, including improved injection system or multiple-pulse injection strategy.
- Accurate spray-wall impingement mechanism, especially the fuel, including alternative fuel, because the physical and chemical characteristics changed with the addition of alternative fuel.
- The effect of the impingement target on mixture formation and emission distribution in the cylinder; in addition, optimization of the impingement target should consider the factors of injection timing, injection angle, and piston structures simultaneously.
- The effects of the early injection parameters on combustion noise radiation, which is expected to gain interest with the development of HCCI and PCCI engines.
- The interrelationship between early injection strategy and modern catalytic devices, such as Diesel Particulate Filter (DPF), Selective Catalytic Reduction (SCR), and Lean NOx Trap (LNT), especially with respect to particle number concentration and distribution, but also possibly due to durability issues.
- Early injection strategy combined with reactivity controlled compression ignition (RCCI), research and analysis of the impact of early injection strategy on RCCI engine, including injection timing, injection pressure, injection angle and so on.
- Combining different types of lubricating oils, one can study the influence of early injection strategies on engine performance and emission characteristics.

**Author Contributions:** X.L. and Z.Z. researched the literatures. H.Z. and Z.Z. analyzed and discussed the results. X.L. and Y.W. provided the future research direction. X.L. and Z.Z. wrote the paper. H.Y. and H.Z. edited the paper.

**Funding:** This research was funded by the National Natural Science Foundation of China, grant number 91641111, 51976135 and 51806148. The author also wants to appreciate the support of Marine Low-Speed Engine Project (Phase I) of MIIT.

**Conflicts of Interest:** The authors declare no conflict of interest.

### Nomenclature

| | |
|---|---|
| HCCI | homogeneous charge compression ignition |
| PCCI | premixed charge compression ignition |
| RCCI | reactivity controlled compression ignition |
| NOx | nitrogen oxides |
| EGR | exhaust gas recycling |
| TDC | top dead center |
| BTDC | before top dead center |
| CO | carbon monoxide |
| HC | hydrocarbon |
| DME | dimethyl ether |
| IMEP | indicated mean effective pressure |
| DI | direct injection |
| HSDI | high-speed direct-injection |
| FIP | fuel injection pressure |
| PM | particulate matter |
| HRR | heat release rate |
| BTE | brake thermal efficiency |
| BSFC | brake-specific fuel consumption |
| ASOI | after start of injection |
| CAD | crank angle degree |
| CI | compression ignition |
| LTR | low-temperature reaction |
| HTR | high-temperature reaction |
| L/HTC | low-/high-temperature combustion |
| ATDC | after top dead center |
| HCHO | formaldehyde |

| | |
|---|---|
| DGB | diesel/gasoline, diesel/n-butanol |
| BSHC | brake specific HC emissions |
| BSCO | brake specific CO emissions |
| BSNOx | brake specific NOx emissions |
| CFD | computational fluid dynamics |
| MIT | main injection timing |
| ISFC | indicated specific fuel consumption |
| CRDI | common rail direct-injection |
| DBE | diesel biodiesel ethanol |
| SOI | start of injection |
| DPF | diesel particulate filter |
| SCR | selective catalytic reduction |
| LNT | lean NOx trap |
| DDCL | diesel from direct coal liquefaction |
| HD | heavy duty |
| ROHR | rate of heat release |
| $P_{max}$ | maximum pressure |

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
