# Peer review of "A Review of Early Injection Strategy in Premixed Combustion Engines"

_applsci, doi:10.3390/app9183737_

Round 1
Reviewer 1 Report
There are many punctuation mistakes in the manoscript.
Reviewer 2 Report
You've presented a detailed synthesis of literature findings related to HCCI vs PCCI. Some of the paragraphs are fairly lengthy but they can be interpreted within two readings. You have also included a lot of figures from previous publications. These add value to the discussion and I believe all of them cite the original reference in the figure captions. What I am uncertain of is whether you have permission to republish these as I am not clear how copyright restrictions apply in this case.
Reviewer 3 Report
This paper presents a review of early injection in diesel- and alternative-fuel engines, with an emphasis on the effect of the injection parameters on performance and emissions. Significant information is extracted from a large number of papers and a list of interesting conclusions are gathered, which allow the reader to have a clear view of advantages and drawbacks of different injection strategies. The work achieved is important and useful.
Yet, the title should mention that this is a review paper, to my opinion. I leave this choice to the authors/editor but I believe that this would enhance the citation potential.
And the presentation of the results could be improved. Proofreading by a native English speaker is recommended, as there are several grammatical and syntax errors. Please find below a non-exhaustive list of language and formatting issues that need addressing.
Figures should be clear enough to be self-contained. Figures 2, 3, 5, 6, 9, 10-13 must be revised. Please check: (i) that all axes are defined - e.g., Figures 2 and 3; (ii) that colormaps are defined or at least described - e.g., Figures 6 and 12; (iii) that all terms, including abbreviations, used within the Figure are defined in the caption.
Present and past tenses are used "randomly" throughout the paper. Please pick up one tense (personally, I believe that the present is nicer to read) and make sure that all the paper is written consistently.
Please provide the expanded form of ALL abbreviations the first time they occur in the paper (excluding the abstract). A list of abbreviations would also be appreciated.
A nomenclature is not required as only a few terms are used. But please define them. For example, φ and T must be defined l.36 p.1.
Please also check inconsistencies between fonts used in Figures and that used in the text. For example, in Figure 1, the local equivalence ratio is referred to with Φ, while φ is used in the text.
In multiple places, "which" is used without a comma (for example, l.11 p.1). As a non-restrictive relative clause, "which" must be preceded by a comma.
In the abstract, "Although" is used incorrectly, l.18 p.1: a main clause is missing. A possibility is to use "However, " instead. (Although is a subordinating conjunction used to connect a subordinate clause to a main clause). Please check that this does not occur in other places of the article.
